

# Efficient urban canopy parametrization for atmospheric modelling: description and application with the COSMO-CLM model (version 5.0_clm6) for a Belgian Summer

Hendrik Wouters[1], Matthias Demuzere[1], Ulrich Blahak[2], Krzysztof Fortuniak[3], Bino Maiheu[4], Johan Camps[5], Daniël Tielemans[6], and Nicole P.M. van Lipzig[1]

[1]KU Leuven, Dept. Earth and Environmental Sciences, Celestijnenlaan 200E, 3001 Heverlee, Belgium
[2]Deutscher Wetterdienst, Frankfurter Strasse 135, 63067 Offenbach, Germany
[3]University of Lodz, Department of Meteorology and Climatology, Narutowicza 88, 90-139 Lodz, Poland
[4]VITO, Flemish Institute for Technological Research, Boeretang 200, 2400 Mol, Belgium
[5]SCK•CEN, Belgian Nuclear Research Centre, Boeretang 200, 2400 Mol, Belgium
[6]VMM, Flemish Environmental Agency, Dokter De Moorstraat 24-26, 9300 Aalst, Belgium

*Correspondence to:* Hendrik Wouters (hendrik.wouters@kuleuven.be)

**Abstract.** This paper presents the Semi-empirical URban-canopY parametrization SURY, which bridges the gap between bulk urban land-surface schemes and explicit-canyon schemes. Based on detailed observational studies, modelling experiments and available parameter inventories, it offers a robust translation of urban canopy parameters containing the three-dimensional information into bulk parameters. It is extremely suitable for an intrinsic representation of canopy-dependent urban physics in existing bulk urban land-surface schemes of atmospheric models. At the same time, it delivers high efficiency in terms of computational cost for long-term climate modelling and numerical weather prediction. SURY enables versatility and consistency in choosing between the urban canopy parameters from bottom-up inventories and bulk parameters from top-down estimates. SURY is tested for Belgium at $2.8\,\mathrm{km}$ resolution with the COSMO-CLM model (version 5.0_clm6) that is extended with the bulk urban land-surface scheme TERRA_URB (version 2). The model reproduces very well the urban heat islands observed from in-situ urban-climate observations, satellite imagery and tower observations, which is in contrast to the original COSMO-CLM model without an urban land-surface scheme. As an application of SURY, the sensitivity of the COSMO-CLM model in terms of land-surface temperatures, air temperatures and associated urban heat islands is quantified for the urban canopy parameter ranges from the Local Climate Zones classification system. On the one hand, their city-scale effect shows that additional urban canopy information has potential for improving regional atmospheric modelling. On the other hand, the model performance and its sensitivity to the different urban canopy parameters largely depend on the temperature quantity considered. Such an ambiguity demonstrates that a multi-variable model evaluation is a requirement for improving and comparing online urban atmospheric modelling strategies.



# 1 Introduction

Cities over the world are expanding (Seto et al., 2012) and an increasing share of the population tends to live in the cities
(United Nations, 2014). The associated changes to the landscape and anthropogenic heating lead to excess temperatures in
cities compared to their natural surroundings, which is known as the urban heat island (UHI) effect. The UHI causes a higher

exposure to heat stress leading to excess mortality Laaidi et al. (2011); Gabriel and Endlicher (2011), damage to infrastructure,
higher energy usage (eg., indoor active cooling) intensifying outdoor urban heat and greenhouse gas emissions, and pressure on
socio-economic activities. In view of global climate change with more extreme heat waves ahead, vulnerabilities and impacts
are increasing for urban centres of all sizes, economic conditions and site characteristics across the world (Revi et al., 2014).

During the past three decades, a vast amount of urban land-surface schemes have been developed. Even though their purpose

of representing urban physics in land-surface schemes of atmospheric models is the same, they differ in terms of modelling
strategy, complexity, input parameters and applicability (see also Grimmond et al., 2011; Best and Grimmond, 2015): On the
one hand, the bulk schemes (eg., Meng, 2015; De Ridder et al., 2015; Wouters et al., 2015; Pleim et al., 2014; Grossman-
Clarke et al., 2005; Fortuniak et al., 2004) take into account the overall radiative, thermal, turbulent-transfer properties, and
water-storage capacity of the urban canopy with a set of bulk parameters. These model parameters are estimated from model

sensitivity experiments (Wouters et al., 2015; De Ridder et al., 2012; Demuzere et al., 2008; Fortuniak, 2005) and observational
campaigns (Masson et al., 2008; Rotach et al., 2005; Offerle et al., 2005). The bulk schemes are suitable for capturing the
general characteristics of the urban climate in regional climate modelling in an efficient way. They generally include the
excess conversion of incoming radiation into sensible heat and the additional anthropogenic heating. As a result, they are
capable of modelling the processes leading to the urban heat island effect. However, they do not explicitly resolve the complex

processes depending on the local characteristics of the urban canopy, which further modulate the urban climate. These processes
include shadowing and multiple scattering of radiation, heterogeneous surface-atmospheric interaction in terms of turbulent
momentum, heat and moisture transport, and the inner-building energy budget. On the other hand, the more complex 'explicit-
canyon' schemes Demuzere et al. (2014); Allegrini et al. (2014); Schubert et al. (2012); Hénon et al. (2012); Oleson et al.
(2008); Fortuniak (2007); Kanda et al. (2005); Martilli et al. (2002); Masson (2000); Crawley et al. (2000) explicitly capture

of one or more of these complex physical processes. As a result, they allow for representing the detailed micro-scale features,
such as heterogeneous temperatures of the facets and canyon wind gusts. Therefore, their added value compared to the less
complex bulk schemes lies in their ability to acquire the more detailed information about urban climate risk and vulnerability
at the micro-scales. Yet, the applicability of these explicit-canyon schemes for atmospheric modelling is sometimes hindered
by either the lack of detailed urban canopy information, computational cost and their model complexity: At first, the urban

canopy parameters, which include information about urban morphology and material properties, are obtained from detailed
inventories (eg., Loridan and Grimmond, 2012; Jackson et al., 2010). The World Urban Database and Access Portal Tools
(http://www.WUDAPT.org) is a recent initiative to collect data on the form and function of cities around the world. Hereby,
Local Climate Zones are identified by means of a decentralized user-base platform employing the urban-classification system
of Stewart and Oke (2012). Despite those initiatives, acquisition and updates of those detailed urban canopy information remain



challenging (Seto et al., 2011), not to mention the scenario prognoses of future parameter changes. Consequently, the spatial detail, accuracy or coverage of the detailed urban canopy parameter datasets are limited, which is also clear from the substantial differences between the datasets (Schneider et al., 2009). For those complex schemes, missing data could deteriorate the model performance (Grimmond et al., 2011). At second, explicit-canyon schemes are computationally demanding compared to bulk schemes. Complex schemes could lead up to $15\%$ percent of total computational cost of an atmospheric model (eg., Trusilova et al., 2015). In contrast, bulk schemes allow very fast downscaling of ensemble climate projections (eg., Lauwaet et al., 2015). Finally, the complexity of the explicit-canyon schemes sometimes makes a consistent incorporation into the host atmospheric model and its maintenance very challenging.

This paper presents the Semi-empirical URban canopY-parametrization SURY (Section 2.1) for bridging the gap between bulk schemes and explicit-canyon schemes. Based on detailed observational studies, modelling experiments and available parameter inventories, it aims for a robust translation of urban canopy parameters containing the three-dimensional information into bulk parameters. In this way, it allows for introducing an intrinsic representation of canopy-dependent urban physics in existing bulk urban land-surface schemes, while preserving their low computational cost.

SURY is implemented in the COSMO(-CLM) model (Section 2.2). The latter is extended with the bulk urban land-surface scheme TERRA_URB version 2.0 which allows for taking bulk parameters from SURY into account. The model system is set up over Belgium during a mid-summer period in 2012 at $2.8\,\mathrm{km}$ resolution (Section 2.3). It is compared with observational data from in-situ urban-climate observations of air temperature and land-surface temperatures from satellite imagery. As shown by Wouters et al. (2013) with an idealized boundary-layer advection model, the nocturnal boundary-layer urban heat island (BLUHI) intensity and its vertical extent, reaching their maximum during the night, are affected by the vertical temperature profile in the lowest few hundreds of metres above the ground. Therefore, an evaluation is also performed against the nocturnal boundary-layer temperatures from tower observations. The meteorological measurements are described in Section 2.4. The COSMO(-CLM) model without an urban land-surface scheme has already been evaluated extensively for numerical weather prediction at the convection-permitting scale (Baldauf et al., 2011) and regional climate modelling applications (Brisson et al., 2016, 2015; Prein et al., 2015; Bucchignani et al., 2015; Thiery et al., 2015; Vanden Broucke et al., 2015; Fosser et al., 2014; Ban et al., 2014; Feldhoff et al., 2014; Dosio et al., 2014; Kotlarski et al., 2014; Lange et al., 2014; Van Weverberg et al., 2014; Panitz et al., 2013; Berg et al., 2012; Keuler et al., 2012, and references herein). Therefore, the evaluation (Section 3.1) focusses on urban-climate modelling, particularly the thermal contrast between cities and the natural surroundings. It is investigated whether the model can reproduce, the canopy-layer UHI (CLUHI; estimated from the urban/rural differences in screen-level temperature), the surface UHI (SUHI; urban/rural differences in the land-surface temperatures) and the nocturnal boundary-layer urban heat island (BLUHI; urban/rural differences in the vertical temperature profiles). As an application of SURY, the effect of urban canopy parameter changes on regional climate modelling with the COSMO(-CLM) model is quantified with an online sensitivity experiment (Section 3.2). The sensitivity allows for addressing the possible effect of the parameters' uncertainty and variability on urban-climate modelling, hence allows for setting priorities in acquiring them. Finally, a discussion and concluding remarks are given in Section 4.





| urban-canopy parameters (input of SURY) | | |
|---|---|---|
| parameter name | symbol | default values |
| substrate albedo | $\alpha$ | 0.101 |
| substrate emissivity | $\epsilon$ | 0.86 |
| substrate heat conductivity | $\lambda_s$ | $0.777\,\mathrm{W\,m^{-1}\,K^{-1}}$ |
| substrate heat capacity | $C_{v,s}$ | $1.25 \times 10^6\,\mathrm{J\,m^{-3}\,K^{-1}}$ |
| building height | $H$ | $15\,\mathrm{m}$ |
| canyon height-to-width ratio | $\frac{H}{W}$ | 1.5 |
| roof fraction | $R$ | 0.667 |
| bulk parameters (output of SURY) | | |
| parameter name | symbol | surface-level values corresponding to urban-canopy defaults |
| bulk albedo | $\alpha_{\mathrm{bulk}}$ | 0.081 (snow-free) |
| bulk emissivity | $\epsilon_{\mathrm{bulk}}$ | 0.89 (snow-free) |
| bulk heat conductivity | $\lambda_{\mathrm{bulk}}$ | $1.55\,\mathrm{W\,m^{-1}\,K^{-1}}$ |
| bulk heat capacity | $C_{v,\mathrm{bulk}}$ | $2.50 \times 10^6\,\mathrm{J\,m^{-3}\,K^{-1}}$ |
| bulk thermal admittance | $\mu_{\mathrm{bulk}}\left(= \sqrt{C_{v,\mathrm{bulk}}\,\lambda_{\mathrm{bulk}}}\right)$ | $1.97 \times 10^3\,\mathrm{J\,m^{-2}K^{-1}s^{-1/2}}$ |
| aerodynamic roughness length | $z_0$ | $1.125\,\mathrm{m}$ |
| inverse Stanton number | $kB^{-1}$ | 13.2 [in case that $u_* = 0.25\,\mathrm{m\,s^{-1}}$] |

**Table 1.** The upper panel shows the urban canopy parameters. They are taken as input for the Semi-empirical URban canopY parametrization (SURY). The default values are adopted from the medium density urban class in Loridan and Grimmond (2012). The lower panel show the bulk parameters, which is the output of SURY. Herbey, $u_*$ refers to the friction velocity.

## 2 Methodology

### 2.1 Semi-empirical urban canopy parametrization

In this section, the Semi-empirical URban canopY parametrization SURY is described. The translation of urban canopy parameters into urban bulk parameters takes into account the urban physical processes with regard to the ground-heat transport

5 (see Section 2.1.1), the surface-radiation exchanges (see Section 2.1.2), and the surface-layer turbulent transport for momentum, heat and moisture (see Section 2.1.3). In this way, SURY introduces an efficient dependency of bulk urban land-surface schemes to the canopy parameters. The robustness of SURY is verified by comparing bulk parameters from top-down estimates with those translated from bottom-up urban canopy parameter inventories. Default values of the urban canopy parameters and those of the translated bulk parameters are determined. An overview of the urban canopy parameters (SURY input) and the

10 bulk parameters (SURY output) is given in Table 1.



### 2.1.1 Ground heat transport

A new methodology is developed for translating the urban canopy parameters into bulk ('effective') thermal parameters for heat capacity and heat conductivity. The latter is taken into account in the 'slab representation', which considers the one-dimensional heat equation of a vertical column commonly used in existing land-surface schemes. In the methodology, the buildings and pavements are considered as massive impermeable structures stacked on the natural soil. It takes into account the three-dimensional top-surface curvature of the urban canopy, which results in a larger contact surface with the atmosphere than a slab surface enhancing the ground heat flux. As denoted by Fortuniak et al. (2004), this results in generally larger urban bulk thermal admittances $\mu_{\text{bulk}} = \sqrt{\lambda_{\text{bulk}} C_{v,\text{bulk}}}$ (De Ridder et al., 2012; Demuzere et al., 2008) than the corresponding substrate values (Loridan and Grimmond, 2012; Jackson et al., 2010). The bulk values of the heat capacity and heat conductivity at the surface level are obtained by both multiplying their substrate counterparts with the Surface-Area Index (SAI) of the roughness elements over land. The latter is the ratio between the land-surface area (excluding vegetation) and the plan area. In this way, the ground heat transport is integrated over the SAI. Such a derivation of the bulk thermal parameters is substantiated as follows: The lateral ground heat transport within through surface substrate is calculated by Fourier's law:

$$Q = -\lambda_s \frac{\partial T}{\partial z} \tag{1}$$

where $\lambda_s$ (units: $[\text{W}\,\text{m}^{-1}\,\text{K}^{-1}]$) is the substrate heat conductivity, and $\frac{\partial T}{\partial z}$ is the vertical temperature gradient of the substrate. At the same time, the tendency of the vertical profile of the surface substrate is described by the heat equation (assuming the substrate heat conductivity independent of ground depth):

$$\frac{\partial T}{\partial t} - \frac{\lambda_s}{C_{v,s}} \frac{\partial^2 T}{\partial z^2} = 0 \tag{2}$$

where $C_{v,s}$ is the substrate heat capacity (units: $[\text{W}\,\text{m}^{-3}\,\text{K}^{-1}]$). We now consider the case that the surface of the substrate in contact with the atmosphere is enlarged with a SAI factor, while conserving the original vertical temperature profile of the substrate. On the one hand, more heat goes through the larger substrate surface, hence the total heat flux through the surface substrate is multiplied with the same factor SAI. In order to get such an enhanced heat transport in the slab representation, $\lambda_s$ needs to be multiplied with SAI as well. On the other hand, the tendency of the vertical temperature profile of the new surface substrate remains the same as the original (this will be the case if the atmospheric forcing would not change, which is true for an infinitesimal time period after enlarging the substrate). For the latter, a heat equation for the slab representation is required that is equivalent to the original equation, ie. Equation 2), hence $\lambda_s/C_{v,s}$-ratio needs to be unchanged. In conclusion, the multiplication of the ground heat flux by a factor SAI and the conservation of the original ground temperature tendency are required in the slab representation. These requirements can be attained with the original set of one-dimensional equations by using bulk parameters for which both $\lambda_s$ and $C_{v,s}$ are multiplied with the same factor SAI. The increase in the bulk thermal parameters due to increased contact surface are taken into account by the ground heat transport module as follows. Firstly, the SAI is calculated. In the case of an urban canopy with parallel streets, SAI can be obtained from the canyon height-to-width





ratio ($\frac{H}{W}$) and the roof fraction $R$ (considering flat roofs):

$$\mathrm{SAI} = \left(1 + 2\frac{H}{W}\right)(1 - R) + R \tag{3}$$

Secondly, the bulk heat capacity at the surface top $C_{v,\mathrm{bulk,s}}$ is obtained by multiplying the heat capacity of the surface substrate $C_{v,s}$ with SAI, which can be done for both the urban canopy and natural land cover:

$$C_{v,\mathrm{bulk,s}} = \mathrm{SAI}\, C_{v,s}. \tag{4}$$

In case of the urban canopy, the latter refers to the heat capacity of urban substrate of the buildings and pavements. Finally, the bulk thermal parameters for the different model levels below the substrate can be determined, which are taken into account by the land-surface module considering a vertical column. Hereby, the natural soil depth is considered, at which the bulk heat capacity $C_{v,\mathrm{bulk}}$ turns into that of the natural soil $C_{v,\mathrm{soil}}$ below that is adopted from the host land-surface model. It is assumed

that this parameter is equal to the averaged height of the roughness elements (excluding vegetation) over land $h$. In the case of the urban canopy, this equals to the building height $h = H$. In between the surface substrate and the depth $h$, a linear transition is hypothesized. This results in the following formulation for the vertical profile of the bulk heat capacity:

$$
\begin{aligned}
C_{v,\mathrm{bulk}}(d) &= \left(1 - \frac{d}{h}\right) C_{v,\mathrm{bulk,s}} + \frac{d}{h} C_{v,\mathrm{soil}}, \text{ for } d < h \\
&= C_{v,\mathrm{soil}}, \text{ for } d \geq h
\end{aligned}
\tag{5}
$$

An analogous formulation is considered for the vertical profile of the bulk ground heat conductivity $\lambda_{\mathrm{bulk}}$:

$$\lambda_{\mathrm{bulk,s}} = \lambda_s\, \mathrm{SAI} \tag{6}$$

where $\lambda_s$ is the substrate heat conductivity, and $\lambda_{\mathrm{bulk,s}}$ is the bulk heat conductivity at the surface top. At a depth $d$ below the surface, the bulk heat conductivity equals to:

$$
\begin{aligned}
\lambda_{\mathrm{bulk}}(d) &= \left(1 - \frac{d}{h}\right) \lambda_{\mathrm{bulk,s}} + \frac{d}{h} \lambda_{\mathrm{soil}}, \text{ for } d < h \\
\lambda_{\mathrm{bulk}}(d) &= \lambda_{\mathrm{soil}}, \text{ for } d \geq h
\end{aligned}
\tag{7}
$$

where $\lambda_{\mathrm{soil}}$ heat conductivity of is the natural soil adopted from the host land-surface model.

Default urban canopy parameters are derived from medium urban density class in Loridan and Grimmond (2012): In case of the urban canopy, the substrate heat conductivity $C_{v,s}$ and substrate heat capacity $\lambda_s$ are set equal to $1.25 \times 10^6\,\mathrm{J\,m^{-3}\,K^{-1}}$ and $0.777\,\mathrm{W\,m^{-1}\,K^{-1}}$, respectively. The height of the roughness elements $h$ (in this case, this is equal to the building height $H$) equals to $15\,\mathrm{m}$. Default values for the $\frac{H}{W}$-ratio and a roof fraction $R$ are set to 1.5 and 0.667, respectively. According to Equation 3, the values for $R$ and $\frac{H}{W}$ lead to an SAI of 2.0. Using the latter results in an surface-level bulk thermal admittance,

expressed as

$$\mu_{\mathrm{bulk,s}} = \sqrt{\lambda_{\mathrm{bulk,s}}\, C_{v,\mathrm{bulk,s}}}. \tag{8}$$

Taking into account the default values for $C_{v,\mathrm{bulk,s}}$ and $\lambda_{\mathrm{bulk,s}}$ above and Equations 4 and 6, it amounts to $1.97 \times 10^3\,\mathrm{J\,m^{-2}\,K^{-1}\,s^{-1/2}}$ for the urban canopy. This value lies within range for the thermal transmittance of the 'compact' and 'open' climate zones in





Table 4 of Stewart and Oke (2012), and also within the uncertainty range obtained by De Ridder et al. (2012). Although this is not a formal validation, these correspondences give confidence to the default substrate parameter values of $C_{v,s}$ and $\lambda_s$ and the enhanced effective heat capacity and heat conductivity for the surface substrate expressed by Equations 4 and 6.

It needs to be noted that the presented methodology above assumes a homogeneous ground temperature of the surface substrate consisting of the different facets in the urban canopy. This is also case for the next section with regard to the surface radiation properties. Consequentially, the scheme does not explicitly represent the temperature variety among the different elements in the urban canopy resulting from shadowing and the heterogeneous thermal and radiative properties. Therefore, urban-physical processes resulting from such variety are not explicitly resolved. This choice was made for providing consistency with the bulk urban land-surface schemes employing bulk parameters.

### 2.1.2 Surface radiation

In this section, the methodology for deriving the bulk (or effective) albedo $\alpha_{\text{bulk}}$ and emissivity $\epsilon_{\text{bulk}}$ from urban canopy parameters is addressed. Hereby, the bulk values refer to the portions of reflected incoming short-wave radiation and emitted infra-red radiation by the urban canopy layer to the upper atmosphere, respectively. It also accounts for the modulation of the bulk value according to the increased-albedo effect of snow. The bulk albedo reduction factor of the urban canopy $\psi_{\text{bulk}}$ is derived from the $\frac{H}{W}$-ratio and roof fraction $R$:

$$\alpha_{\text{bulk}} \simeq ((1 - f_{\text{snow}})\alpha + f_{\text{snow}}\,\alpha_{\text{snow}})\,\psi_{\text{bulk}}\left(\frac{H}{W}, R\right) \tag{9}$$

where $\alpha$ is the substrate albedo and $f_{\text{snow}}$ is the snow-covered fraction. Hereby, $\psi_{\text{bulk}}\left(\frac{H}{W}, R\right)$ is calculated by:

$$\psi_{\text{bulk}}\left(\frac{H}{W}, R\right) = R + (1 - R)\,\psi_{\text{canyon}}(\frac{H}{W}) \tag{10}$$

where $\psi_{\text{canyon}}(\frac{H}{W})$ is the canyon albedo reduction factor. Instead of implementing a computationally demanding explicit canyon radiation scheme, an approximation for $\psi_{\text{canyon}}$ is proposed to the numerical estimation from Fortuniak (2007). The latter applies an exact solution of the multiple-reflection problem allowing to subdivide the different facets in an urban canyon. The exact solution results in a high accuracy for low solar heights when the lower canyon parts are shaded. It could reproduce the effective-albedo observations from a scale model Aida (1982) and from a real canyon very well. The numerical estimation shows that the albedo reduction is the most sensitive to the $\frac{H}{W}$-ratio, hence the following approximation is proposed:

$$\psi_{\text{canyon}}\left(\frac{H}{W}\right) = \exp\left(-0.6\,\frac{H}{W}\right) \tag{11}$$

This closely matches the numerical estimation with a maximal error of $\pm 7\%$ for the highest excursion of the sun during summer solstice at the mid-latitude ($55°$), a canyon parallel to the solar azimuth, and a substrate albedo of 0.4 (Fortuniak, 2007, see their Figure 11). With regard to other sun heights, canyon directions relative to the solar azimuth, and $\frac{H}{W}$-ratios between 0 and 2 (Fortuniak, 2007, see their Figures 8 and 11), the proposed $\psi_{\text{canyon}}$-formulation has a maximal error of $45\%$. It should be noted that the approximation is fitted to the numerical estimation for a perfect urban canyon. Hence, the approximation neglects additional albedo changes due to bending roofs and varying albedos for the different facets.



Optionally, a distinction is made between the albedo of roofs, roads and walls as follows:

$$\alpha_{\text{bulk}} \simeq \frac{\left[ \alpha_{\text{street,snow}} + 2\frac{H}{W}\alpha_{\text{wall,snow}} \right]}{(1 + 2\frac{H}{W})} \psi_{\text{canyon}} \left( \frac{H}{W} \right)(1 - R) + \alpha_{\text{roof,snow}} R \tag{12}$$

with

$$\alpha_{i,\text{snow}} = (1 - f_{\text{snow}})\alpha_i + f_{\text{snow}}\alpha_{\text{snow}}, \text{ for } i \text{ in (roof, wall, street)} \tag{13}$$

and where $\frac{\left[ \alpha_{\text{street,snow}} + 2\frac{H}{W}\alpha_{\text{wall,snow}} \right]}{(1 + 2\frac{H}{W})}$ is the averaged substrate albedo of the street and wall surfaces in the urban canyon. The effective infra-red emissivity $\epsilon_{\text{bulk}}$ takes into account the same bulk albedo reduction factor $\psi_{\text{bulk}}$ as follows:

$$\epsilon_{\text{bulk}} = 1 - \psi_{\text{bulk}}\left( 1 - ((1 - f_{\text{snow}})\epsilon + f_{\text{snow}}\epsilon_{\text{snow}}) \right) \tag{14}$$

where $\epsilon$ is the substrate emissivity and $\epsilon_{\text{snow}}$ is the snow emissivity.

The robustness of the Equations 11, 12 and 13 is verified for a dense urban area in Toulouse centre: The averaged substrate albedo for walls, roofs and roads are 0.25, 0.15 and 0.08, respectively, whereas the roof fraction and $\frac{H}{W}$-ratio are 0.59 and 1.4 respectively, see Pigeon et al. (2008). This yields a snow-free effective albedo $\alpha_{\text{bulk}}$ for the urban canopy of 0.125. This is close to the bulk value of 0.13 for Summer, which is estimated from the averaged ratio between the upward and the downward radiation measured by a mast tower during the 'CAPITOUL'-campaign (Masson et al., 2008).

Default values for substrate albedo of roofs (0.10), walls (0.10) and roads (0.15) are adopted from (Loridan and Grimmond, 2012). Regarding the default values for $\frac{H}{W}$ and $R$ of 1.5 and 0.67, Equation 12 yields a (snow-free) effective albedo for the urban canopy of $\alpha_{\text{bulk}} = 0.081$. Given the values for $R$ and $\frac{H}{W}$, the bulk albedo reduction factor for the urban canopy yields $\psi_{\text{bulk}} = 0.80$. Together with the more simple formulation Equation 9, the value for the substrate albedo for $\alpha$ of 0.10 is obtained and used by default. Analogously, the default value is obtained for the snow-free effective emissivity of 0.89 and the substrate emissivity of 0.86.

### 2.1.3 Surface-layer turbulent transport

Following Sarkar and De Ridder (2010), the aerodynamic roughness lengths for the urban canopy is calculated as follows:

$$z_0 = 0.075H \tag{15}$$

with $H$ the building height. The thermal roughness length $z_{0\text{H}}$ is obtained with a parametrization of the inverse Stanton number (as in De Ridder, 2006; Demuzere et al., 2008):

$$kB^{-1} = \ln \left( \frac{z_0}{z_{0\text{H}}} \right) \tag{16}$$

with $k$ the von Kàrmàn constant. For the urban canopy, a bluff-body thermal roughness length parametrization from Brutsaert (1982) is introduced using parameter values from Kanda et al. (2007):

$$kB^{-1} = 1.29 Re_*^{0.25} - 2.0 \tag{17}$$

where $Re_* = u_* z_0 / \nu$ is the roughness Reynolds number, $u_*$ is the friction velocity, and $\nu = 1.461 \times 10^{-5} \, \text{m}^2 \, \text{s}^{-1}$ the kinematic viscosity of air.





## 2.2 The COSMO(-CLM) model

The COSMO model (Steppeler et al., 2003) is a full-3D atmospheric numerical model designed for operational and research applications in limited-area weather prediction at high resolution. The model has been developed by the German Weather Service (DWD) and is further improved and maintained by the Consortium for Small-Scale Modelling (COSMO). Members of this consortium include several meteorological services from inside and outside Europe. The COSMO model has a compressible non-hydrostatic core for atmospheric dynamics, and includes parametrizations for radiative transfer, cloud microphysics, subgrid-scale turbulence and convection. It also includes parametrizations for the ground heat- and water transport and the land-atmosphere interactions, as described in more detail in the next paragraph. The regional climate model COSMO-CLM (COSMO model in CLimate Mode) is based on the COSMO model, and includes modifications allowing the application on time scales up to centuries (Böhm et al. 2006; Rockel et al., 2008). These modifications comprise the introduction of an annual cycle to vegetation parameters like the vegetation cover and the leaf area index as well as an externally prescribed, time-dependent $CO_2$ concentration in the atmosphere. The COSMO-CLM model is further developed by a vast amount of researchers of the CLM-community in- and outside of Europe (http://www.clm-community.eu). It is used extensively for long-term regional-climate studies (Klutse et al., 2015; Endris et al., 2015; Vanden Broucke et al., 2015; Cavicchia et al., 2014; Davin et al., 2014; Schubert and Grossman-Clarke, 2013) and for downscaling global-climate realizations (Akkermans et al., 2014; Dosio and Panitz, 2015; Lejeune et al., 2014).

The ground heat- and water transport and the representation of vegetation and snow-cover are resolved by the Soil-Vegetation-Atmosphere Transfer (SVAT) module TERRA_ML (Schulz et al., 2016; Doms et al., 2011; Grasselt, 2008). The COSMO(-CLM) model implements the next generation TKE-based surface-layer transfer scheme (Doms et al., 2011; Buzzi, 2008). The surface layer, which refers to the layer of air between the earth surface and the lowest model level, is divided into a laminar-turbulent sublayer, the roughness layer, and a constant-flux (or Prandtl) layer. The surface layer scheme is also intimately related to the TKE-based closures of the COSMO(-CLM) model, see sections 3.3 and 3.4 of Doms et al. (2011). As a result, the surface layer does not need the empirical Monin-Obukhov stability functions (as in Paulson, 1970; Guo and Zhang, 2007) for which the Obukhov stability parameter needs to be determined from an iterative procedure or a non-iterative approximation (eg., Louis, 1979; Wouters et al., 2012; Li et al., 2014, and references herein). It rather generates these functions by the use of the dimensionless coefficients of the turbulence closure (Mellor and Yamada, 1982), see section 4.2 of Doms et al. (2011). Land-surface parameters including the soil type, vegetation, and orography are specified with the External Parater tool (EXTPAR), see also Smiatek et al. (2008). These are processed from global land-cover data sets, such as those for orography and coastlines from 'Global 30 Arc-Second Elevation' (GTOPO30; see https://lta.cr.usgs.gov/GTOPO30) or ASTER (Three Arc-Second Elevation dataset), soil data from the 'Digital Soil Map of the World' (DSMW; see http://data.fao.org/map?entryId=446ed430-8383-11db-b9b2-000d939bc5d8), and land-use data from 'Global Land Cover 2000' (GLC2000, see Bartholomé and Belward, 2005), GLOBCOVER (Loveland et al., 2010) or ECOCLIMAP (Faroux et al., 2013). The vegetation parameters, which include vegetation cover fraction, LAI and rooting depth, are specified with annual minimum and maximum values depending on the land-use Doms et al. (2011, see their tables 14.3, 14.4, and 14.5). For the extra-tropical





northern hemisphere, a growing and resting period is calculated according to latitude. The roughness length parameter (over land) depends on both land-use and the subgrid-scale orography. The obtained vegetation and soil data are assigned to the natural soil fraction. The methodology for the reduction in vegetation abundancy in urban environments compared to the rural surroundings according to the underlying land-use dataset.

In the original COSMO(-CLM) model, cities are represented by natural land surfaces with an increased surface roughness length and a reduced vegetation cover. However in this representation, urban areas are still treated as water-permeable soil with aerodynamic, radiative and thermal parameters similar to the surrounding natural land. Therefore, this basic representation could not reliably capture the urban physics and associated urban-climatic effects including urban heat islands. In order to tackle this issue, the bulk scheme TERRA_URB Wouters et al. (2015) has been introduced for providing an intrinsic representation

of urban physics in the COSMO(-CLM) model. In addition to the modified ground heat and moisture transport and surface-atmosphere interactions in urban areas, it also features an impervious water-storage parametrization based on a probability density function of water puddles. The initial release of TERRA_URB (version 1.0) has been evaluated in offline mode in Wouters et al. (2015) for intensive observation campaigns in Basel (Rotach et al., 2005) and Toulouse (Masson et al., 2008). It has also been employed for acquiring heat-stress scenarios of future climate change and urban land-use change scenarios in

Belgium (De Ridder et al., 2015) adopted by the Climate Report of the Flemish Environmental Agency (Brouwers et al., 2015). During the Online Urban Model Intercomparison Project (Trusilova et al., 2015), TERRA_URB v.1 has been compared to other urban land-surface parametrizations coupled to same COSMO-CLM model Trusilova et al. (2013); Schubert et al. (2012). The next version (version 2.0) of TERRA_URB introduces several advancements compared to its previous version. The main advancements are the implementation of SURY and the application of the TKE-based surface-layer turbulent transfer module

of the COSMO(-CLM) model (Doms et al., 2011; Buzzi, 2008). A full description of TERRA_URB version 2.0 can be found in Appendix A.

### 2.3   Model setup

The COSMO(-CLM) model that implements SURY in its urban land-surface module TERRA_URB version 2.0 is setup for the reference simulation over Belgium, see Figure 1. The simulation is performed during a summer period from 2012/7/01 until

2012/08/20 for which the first three weeks are considered as spin-up. The model parameter setup is based on the COSMO-CLM model configurations of Brisson et al. (2016, 2015) and Prein et al. (2015) employing convection resolving climate simulations. The domain covers 175 by 175 grid cells centred over Brussels with a horizontal grid spacing of $2.8\,\mathrm{km}$ resolution. Forty vertical layers are used with the lowest domain level at $10\,\mathrm{m}$ above the ground. For the lateral boundaries, the model takes 3-hourly analysis from the operational model of the European Centre for Medium Range Weather Forecasting (ECMWF) at a

spatial resolution of $0.125°$ in latitude and longitude. The reference simulation above, referred to as 'REF', is compared with a simulation 'STD' that uses the original COSMO-CLM model without urban land-surface parametrization.

    In addition to the REF and STD setup described above, a range of online-coupled sensitivity experiments are performed. Starting from from REF for each sensitivity simulations, the parameters for the urban canopy are changed according to the minimum ('L'ow scenarios) and maximum values ('H'igh scenarios) of the urban canopy parameter ranges taken from the





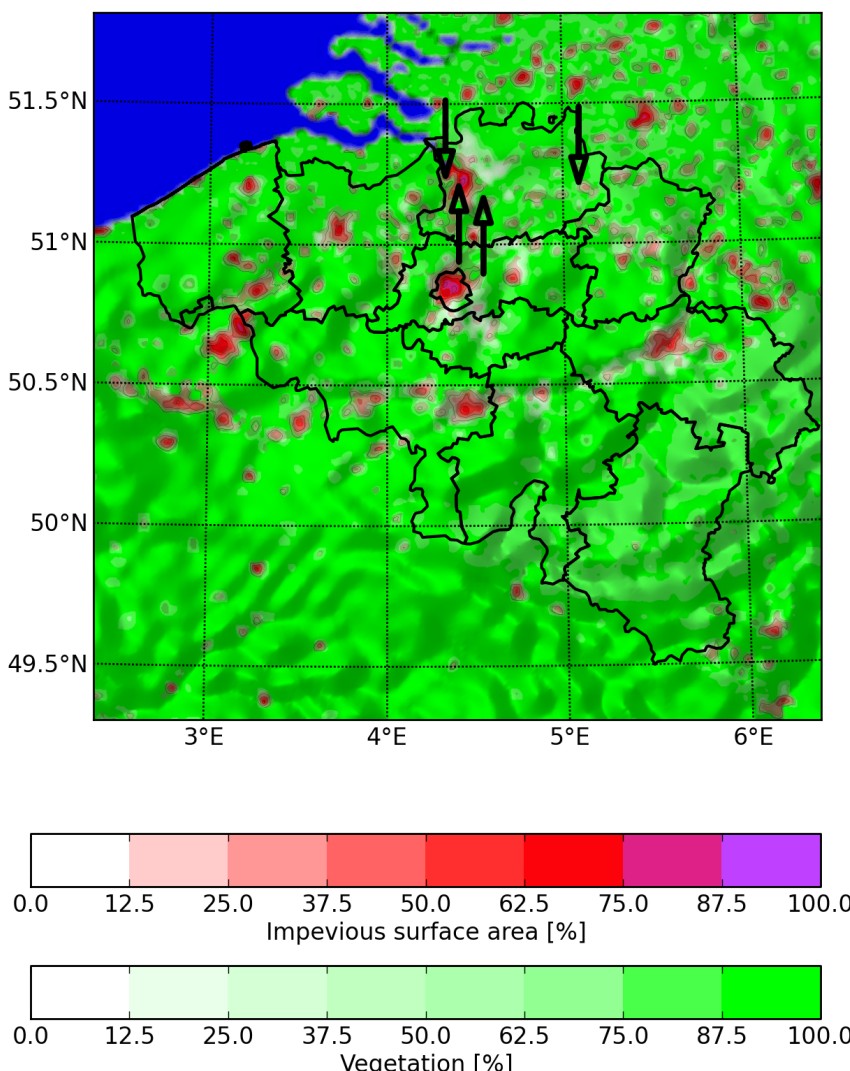

**Figure 1.** Domain composite of the impervious surface area, vegetation cover and orography (indicated with the shading) for the reference (REF) model setup of the COSMO-CLM model coupled to TERRA_URB version 2.0 at 2.8 km resolution over Belgium. The arrows directed upwards indicate the locations of the in-situ observations located in the urban area of Antwerp (left; Royal Lyceum of Antwerp) and in the rural area of Vremde (right; Organic Farm van Leemputten). The arrows directed downwards indicate the locations of the tower observations located at a flat industrial terrain in Zwijndrecht (left arrow) of the Flemish Environmental Agency), and at an rural area (right arrow) of the Belgian Nuclear Research Center (SCK•CEN) in Mol.



| EXP-ID | urban parameter | symbol | L | H |
|---|---|---|---|---|
| A | substrate albedo | $\alpha$ | 0.10 | 0.25 |
| B | substrate heat conductivity | $\lambda_s$ [W m$^{-1}$ K$^{-1}$] | 0.200 | 0.968 |
| C | substrate heat capacity | $C_{v,s}$ [$1 \times 10^6$ J m$^{-2}$ K$^{-1}$] | 0.321 | 1.56 |
| D | canyon height-to-width ratio | $\frac{H}{W}$ | 0.75 | 2.0 |
| E | building height | $H$ [m] | 3 | 30 |
| F | roof fraction | $R$ | 0.40 | 0.70 |
| G | anthropogenic heat emission | AHE | 0 | $2 \times$ FL09 |

**Table 2.** Overview of parameter sensitivity experiments. Seven couples of experiments (AL, AH, BL, BH, CL, CH, DL, DH, EL, EH, FL, FH, FL, GH) are performed for which the default urban canopy parameters are modified to the values in the 'L'ow and 'H'high column. Except for the anthropogenic heat emission (AHE), L and H correspond to the minimum and maximum values of the urban canopy parameter ranges for the local climate zones of compact 'low-rise and mid-rise' defined in Stewart and Oke (2012). For the GL scenario, the AHE is set to $0\,\mathrm{W\,m^{-2}}$. For the GH scenario, AHE multiplied by 2 compared to the default setup for which the dataset and methodology of Flanner (2009, denoted as FL09 in the table).

Local Climate Zones of compact 'low-rise and mid-rise' defined in Stewart and Oke (2012). The parameters are respectively the substrate albedo ($\alpha$), surface heat conductivity ($\lambda_s$), surface heat capacity ($C_{v,s}$), height-to-width ratio ($\frac{H}{W}$), building height ($H$) and roof fraction ($R$). Hereby, the parameter ranges for the heat capacity and heat conductivity are obtained from Equation 8 and employ the range in thermal admittance from Stewart and Oke (2012). In the same way, the parameter range for the substrate albedo is obtained from Equation 9 by considering the urban albedo range $\alpha_{\mathrm{bulk}}$ from Stewart and Oke (2012). In addition, the anthropogenic heat emission (AHE) is set to $0\,\mathrm{W\,m^{-2}}$ in the L scenario, and whereas it is multiplied by 2 in the H scenario. These two scenarios are in agreement with the given uncertainty range in Stewart and Oke (2012) - indicating a range between 0 and $75\,\mathrm{W\,m^{-2}}$ - while preserving daily and annual variability in the model, and spatial variability from the input data. As a result, 14 additional simulations are performed for which the L scenarios are respectively AL, BL, CL, DL, EL and FL, GL, and the H scenarios are AH, BH, CH, DH, EH, FH and GH. An overview of the simulations can be found in Table 2.

## 2.4 Evaluation data

### 2.4.1 In-situ measurements

The modelled screen-level air temperature and the associated CLUHI is evaluated against in-situ measurements for an urban and rural site in Antwerp. These were performed using platinum resistance thermometers supplied by Campbell Scientific. The sensors were mounted in an actively ventilated radiation shield (Young 43503) to reduce heating effects by radiation loading on the sensor and stagnant air inside the shield. The sensor + radiation shield setups were deployed side by side during a test phase during one week at the end of june 2012. The Root-Mean-Square difference on the 15 minute temperature averages during this week was found lower than $0.04\,^{\circ}\mathrm{C}$. The rural station was located on the premises of an organic farm enterprise



some $10\,\text{km}$ to the south east of Antwerp. The station is situated in pasture land with grass kept short by sheep. The nearest buildings are low rise and about $200\,\text{m}$ away. The urban station was located on the premises of the Royal Lyceum in Antwerp, a secondary school. The sensor was mounted in the urban canopy layer on top of the roof of a $3\,\text{m}$ high small building in the centre of a fairly large playground. The distance to the nearest adjacent wall is about $15\,\text{m}$. Though the location on top the roof

is probably not ideal in terms of micro-scale effects, comparisons with similar measurements in the vicinity of the lyceum and an analysis of the urban-rural difference suggest that the measurements at the lyceum can be considered representative for the neighbourhood. The position of the in-situ measurements are indicated in Figure 1.

### 2.4.2 MODIS satellite imagery

As demonstrated recently by Hu et al. (2014) and Tomlinson et al. (2012), satellite information can be used for evaluating

the models' excess surface heating in urban areas compared to the natural surrounding areas for the entire study domain. The model results are evaluated against land-surface temperatures (LSTs) derived from the MODIS sensor on board of the Terra and Aqua satellite. Providing four overpasses a day with a spatial resolution of approximately $1\,\text{km}$. Hu et al. (2014) compared three methods to evaluate MODIS LST to HRLDAS surface temperatures and the optimal technique is employed for this study. For each Terra and Aqua overpass , the nearest model time is selected resulting in images at 9 and 11 am and 20 and 00 pm. The

first two are combined as "day", while the latter two are referred to as "night". Cloudy pixels are removed in MOD11A1 and MYD11A1 Version 5 (Wan, 2008) while potential remaining cloudy pixels are removed by only using MODIS LST data within 1.5 times the interquartile range (Hu et al., 2014; Monaghan et al., 2014). In addition, cloudy pixels from the COSMO-CLM model are removed in both datasets. Finally, in order to minimize view-angle biased MODIS' LSTs while keeping sample sizes larges enough, only overpasses between $-45°$ and $45°$ from nadir are used. All remaining pixels therefore represent clear-sky

conditions and are used to evaluate LST of the COSMO-CLM model.

### 2.4.3 Tower observations

In order to evaluate the nocturnal boundary-layer temperature and BLUHI in the model, observations from two meteorological towers within the province of Antwerp (Belgium) are used. The first tower of the Flemish Environmental Agency (VMM) is $160\,\text{m}$ high and located on an industrial site in Zwijndrecht (Geographical coordinates: 51°14'37.9"N 4°20'3.1"E). Mea-

25 surements for temperature, humidity, wind speed, pressure and precipitation are performed at $8\,\text{m}$, $24\,\text{m}$, $48\,\text{m}$, $80$ , $114\,\text{m}$ and $153\,\text{m}$. The temperature measurements are obtained with multi-stage solid state thermistor (Met One 062). The second tower of the Belgian Nuclear Research Centre (SCK•CEN) is $120\,\text{m}$ high and located in Mol (Geographical coordinates: 51°13'04"N and 5°05'24"E). Measurements have been done in the framework of determining atmospheric stability according to the turbulence scheme of SCK•CEN (Bultynck and Malet, 1972). It is placed in a rural area on flat terrain with pine trees

in the immediate vicinity. Measurements for wind direction, wind speed and temperatures are performed at $8\,\text{m}$, $24\,\text{m}$, $48\,\text{m}$, $69\,\text{m}$, $78\,\text{m}$ and $114\,\text{m}$) above ground level. Hereby, the temperature measurements are obtained with a Cu-Constantan thermocouples The temperatuere measurements of both towers are placed within a ventilator driven aspirated radiation shield to protect against direct and diffuse solar heating. They are continuously recorded on a 1 minute basis in order to make and store



10-minute averages. For the model evaluation, the available temperature measurements at the heights $8\,\mathrm{m}$, $48\,\mathrm{m}$ and $114\,\mathrm{m}$) of both towers have been have been used. The position of the tower observations are indicated in Figure 1.

## 3 Results

### 3.1 Evaluation

The averaged LST and screen-level temperatures for the day- and night-time from the REF simulation during the summer evaluation period are shown in Figure 2 . It is found that the SUHI reaching its maximum during the day is typically larger (approximately $4\,\mathrm{K}$ for the urban centres) than the CLUHI reaching its maximum during the night (approximately $2\,\mathrm{K}$). The night-time CLUHI is of comparable magnitude as the night-time SUHI. The CLUHI reaches a its minimum at noon. Hereby, the urban heat islands occur at the scale of the cities and magnitude increases with city size. These findings are consistent

with existing literature for urban-climate modelling and observational studies (eg., Phelan et al., 2015). Hereby, excess surface heating is taking place in cities at day-time. The excess heat is stored into the urban canopy which leads to day-time excess land-surface temperatures. Subsequently, a prolongated excess heat during the night release takes place from the urban canopy leading to the excess urban screen-level temperatures at night-time. Modeled temperatures and associated urban heat islands of the REF simulation (with parametrization) are evaluated in more detail below against screen-level temperatures from the in-situ

measurements (Section 3.1.1), land-surface temperatures (LST) from satellite imagery (Section 3.1.2) and nocturnal boundary-layer temperature from tower observations (Section 3.1.3). Herein, a comparison is also made with the STD simulation that excludes the urban parametrization.

### 3.1.1 Two-metre air temperatures

The model evaluation of the screen-level air temperature and the CLUHI for Antwerp is displayed in Figure 3 and in Tables 3,

4 and 5. Herein, sensitivity results are also included, which will be addressed in Section 3.2.

For the urban site, reference model REF agrees well with the observed temperatures with a mean bias (MB) of $0.39\,\mathrm{K}$ and a Pearson correlation coefficient (R2) of $0.96$. In contrast, the standard model STD results in a general temperature underestimation with a MB of $-0.95\,\mathrm{K}$ and an R2 of $0.95$. For the rural site, the temperature timeseries, daily variability and diurnal cycle are also well reproduced for both REF and STD with an MB $0.56\,\mathrm{K}$ and $0.14\,\mathrm{K}$ and an R2 of both $0.95$, respectively.

The overestimation for the rural site is larger for REF than for STD, because of the advection of excess heat from urban areas towards the rural areas. The general positive temperature bias of the model for the rural site (for summer) is also found in other evaluations of the COSMO-CLM model (Brisson et al., 2016; Trusilova et al., 2015; Brisson et al., 2015; Lange et al., 2014; Keuler et al., 2012).

The CLUHI intensity ($1.74\,\mathrm{K}$), calculated from the observed difference between the urban station and the rural station, is

30 well reproduced by REF ($1.56\,\mathrm{K}$) with a very good R2 of $0.80$ and a bias of $-0.18\,\mathrm{K}$. In contrast, the CLUHI could not be reproduced by the STD model: With an averaged UHI value of $0.64\,\mathrm{K}$, STD has a much larger model bias $-1.10\,\mathrm{K}$ and a lower





| Royal Lyceum of Antwerp (urban) | | | |
|---|---|---|---|
| EXP-ID | L | H | \|H - L\| |
| OBS | 293.85 ( - ) | - | - |
| STD/REF | 292.90 (-0.95) | 294.24 (+0.39) | 1.34 |
| A | 294.24 (+0.39) | 294.11 (+0.26) | 0.13 |
| B | 294.10 (+0.25) | 294.28 (+0.43) | 0.18 |
| C | 294.08 (+0.23) | 294.27 (+0.42) | 0.19 |
| D | 294.12 (+0.27) | 294.31 (+0.46) | 0.19 |
| E | 294.17 (+0.32) | 294.33 (+0.48) | 0.16 |
| F | 294.38 (+0.53) | 294.24 (+0.39) | 0.14 |
| G | 293.53 (-0.32) | 294.77 (+0.92) | 1.24 |
| Organic Farm Van Leemputten (rural) | | | |
| EXP-ID | L | H | \|H - L\| |
| OBS | 292.12 ( - ) | - | - |
| STD/REF | 292.26 (+0.14) | 292.68 (+0.56) | 0.42 |
| A | 292.68 (+0.56) | 292.71 (+0.59) | 0.03 |
| B | 292.66 (+0.54) | 292.73 (+0.61) | 0.07 |
| C | 292.65 (+0.53) | 292.71 (+0.59) | 0.06 |
| D | 292.67 (+0.55) | 292.71 (+0.59) | 0.04 |
| E | 292.68 (+0.56) | 292.72 (+0.60) | 0.04 |
| F | 292.73 (+0.61) | 292.67 (+0.55) | 0.06 |
| G | 292.31 (+0.19) | 293.01 (+0.89) | 0.70 |
| Difference (urban heat island) | | | |
| EXP-ID | L | H | \|H - L\| |
| OBS | 1.74 ( - ) | - | - |
| STD/REF | 0.64 (-1.10) | 1.56 (-0.18) | 0.92 |
| A | 1.56 (-0.18) | 1.41 (-0.33) | 0.15 |
| B | 1.44 (-0.30) | 1.55 (-0.19) | 0.11 |
| C | 1.43 (-0.31) | 1.56 (-0.18) | 0.13 |
| D | 1.44 (-0.30) | 1.60 (-0.14) | 0.16 |
| E | 1.48 (-0.26) | 1.61 (-0.13) | 0.13 |
| F | 1.65 (-0.09) | 1.57 (-0.17) | 0.08 |
| G | 1.22 (-0.52) | 1.77 (+0.03) | 0.55 |

**Table 3.** Daily averages for modelled and observed temerpatures (unit: [K]) during mid-summer (2012/07/21 until 2012/08/20) at the Royal Lyceum for Antwerp (Urban), the Organic Farm Van Leemputten (Rural), and their difference (the Urban heat island effect of Antwerp). Each first row shows results of the observations. Each second row shows results for COSMO-CLM model without urban parametrization (L column), for COSMO-CLM + TERRA_URB version 2.0 (H column) and their absolute difference ($|H - L|$ column). The remainder rows show each of the urban-parameter sensitivity simulations in Table 2. Hereby, the Low scenarios are shown in the L column, the High scenarios in the H column and the parameter sensitivity Range (absolute difference between L and H) in the $|H - L|$ column.





| Royal Lyceum of Antwerp (urban) | | | |
|---|---|---|---|
| EXP-ID | L | H | \|H - L\| |
| OBS | 292.27 ( - ) | - | - |
| STD/REF | 291.33 (-0.94) | 293.21 (+0.94) | 1.88 |
| A | 293.21 (+0.94) | 293.12 (+0.85) | 0.09 |
| B | 292.81 (+0.54) | 293.28 (+1.01) | 0.47 |
| C | 292.72 (+0.45) | 293.29 (+1.02) | 0.57 |
| D | 292.99 (+0.72) | 293.32 (+1.05) | 0.33 |
| E | 293.10 (+0.83) | 293.34 (+1.07) | 0.24 |
| F | 293.46 (+1.19) | 293.19 (+0.92) | 0.27 |
| G | 292.39 (+0.12) | 293.79 (+1.52) | 1.40 |
| Organic Farm Van Leemputten (rural) | | | |
| EXP-ID | L | H | \|H - L\| |
| OBS | 289.60 ( - ) | - | - |
| STD/REF | 290.55 (+0.95) | 291.05 (+1.45) | 0.50 |
| A | 291.05 (+1.45) | 291.05 (+1.45) | 0.00 |
| B | 290.95 (+1.35) | 291.08 (+1.48) | 0.13 |
| C | 290.91 (+1.31) | 291.09 (+1.49) | 0.18 |
| D | 290.98 (+1.38) | 291.09 (+1.49) | 0.11 |
| E | 291.05 (+1.45) | 291.10 (+1.50) | 0.05 |
| F | 291.13 (+1.53) | 291.04 (+1.44) | 0.09 |
| G | 290.68 (+1.08) | 291.39 (+1.79) | 0.71 |
| Difference (urban heat island) | | | |
| EXP-ID | L | H | \|H - L\| |
| OBS | 2.67 ( - ) | - | - |
| STD/REF | 0.78 (-1.89) | 2.16 (-0.51) | 1.38 |
| A | 2.16 (-0.51) | 2.07 (-0.60) | 0.09 |
| B | 1.87 (-0.80) | 2.21 (-0.46) | 0.34 |
| C | 1.81 (-0.86) | 2.20 (-0.47) | 0.39 |
| D | 2.01 (-0.66) | 2.24 (-0.43) | 0.23 |
| E | 2.05 (-0.62) | 2.25 (-0.42) | 0.20 |
| F | 2.33 (-0.34) | 2.15 (-0.52) | 0.18 |
| G | 1.71 (-0.96) | 2.41 (-0.26) | 0.70 |

**Table 4.** Idem as Table 3, but averaged values for the night (between 6:00PM and 5:59AM UTC).





| Royal Lyceum of Antwerp (urban) | | | |
|---|---|---|---|
| EXP-ID | L | H | \|H - L\| |
| OBS | 295.44 ( - ) | - | - |
| STD/REF | 294.47 (-0.97) | 295.27 (-0.17) | 0.80 |
| A | 295.27 (-0.17) | 295.10 (-0.34) | 0.17 |
| B | 295.40 (-0.04) | 295.27 (-0.17) | 0.13 |
| C | 295.44 (+0.00) | 295.26 (-0.18) | 0.18 |
| D | 295.25 (-0.19) | 295.29 (-0.15) | 0.04 |
| E | 295.23 (-0.21) | 295.32 (-0.12) | 0.09 |
| F | 295.30 (-0.14) | 295.28 (-0.16) | 0.02 |
| G | 294.67 (-0.77) | 295.75 (+0.31) | 1.08 |

| Organic Farm Van Leemputten (rural) | | | |
|---|---|---|---|
| EXP-ID | L | H | \|H - L\| |
| OBS | 294.64 ( - ) | - | - |
| STD/REF | 293.98 (-0.66) | 294.32 (-0.32) | 0.34 |
| A | 294.32 (-0.32) | 294.36 (-0.28) | 0.04 |
| B | 294.38 (-0.26) | 294.38 (-0.26) | 0.00 |
| C | 294.40 (-0.24) | 294.34 (-0.30) | 0.06 |
| D | 294.37 (-0.27) | 294.33 (-0.31) | 0.04 |
| E | 294.31 (-0.33) | 294.34 (-0.30) | 0.03 |
| F | 294.34 (-0.30) | 294.30 (-0.34) | 0.04 |
| G | 293.95 (-0.69) | 294.63 (-0.01) | 0.68 |

| Difference (urban heat island) | | | |
|---|---|---|---|
| EXP-ID | L | H | \|H - L\| |
| OBS | 0.80 ( - ) | - | - |
| STD/REF | 0.49 (-0.31) | 0.96 (+0.16) | 0.47 |
| A | 0.96 (+0.16) | 0.74 (-0.06) | 0.22 |
| B | 1.02 (+0.22) | 0.89 (+0.09) | 0.13 |
| C | 1.05 (+0.25) | 0.92 (+0.12) | 0.13 |
| D | 0.88 (+0.08) | 0.96 (+0.16) | 0.08 |
| E | 0.92 (+0.12) | 0.98 (+0.18) | 0.06 |
| F | 0.96 (+0.16) | 0.98 (+0.18) | 0.02 |
| G | 0.72 (-0.08) | 1.12 (+0.32) | 0.40 |

**Table 5.** Idem as Table 3, but averaged values for the day (between 6:00AM and 5:59PM UTC).

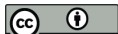



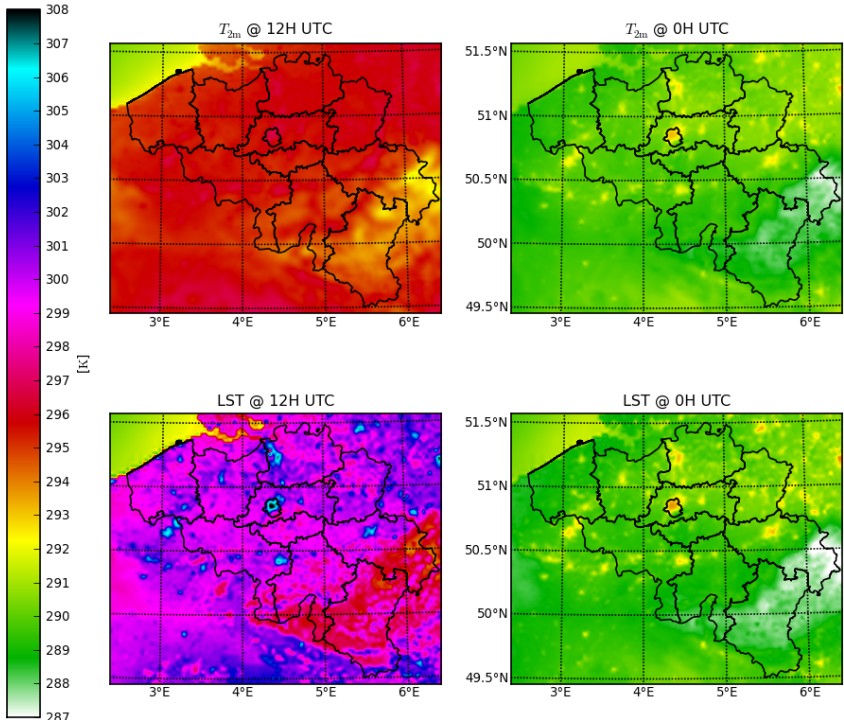

**Figure 2.** Modelled horizontal profiles of screen-level (top panels) and surface (bottom panels) temperatures at noon (left panels) and midnight (right panels), averaged for 2012/07/21 until 2012/08/20.

R2 of 0.54. Again for the REF simulation, an underestimation of the UHI occurs during the night and a slight overestimation in the late afternoon. The modelled standard deviation of $1.25\,\mathrm{K}$ is lower than the observed $1.56\,\mathrm{K}$, which originates from the underestimation of the nocturnal UHI.

### 3.1.2 Land-surface temperatures

5 The model is evaluated with LSTs derived from the MODIS satellite described in Section 2.4.2. Hereby, four urban classes are distinguished: no urban (NU) with Impervious Surface Area (ISA) $\leq 0.05$; light urban (LU) , $0.05 < \mathrm{ISA} \leq 0.25$); medium urban (MU), $0.25 < \mathrm{ISA} \leq 0.5$; dense urban (DU), ISA $> 0.5$. The results are displayed in Figure 4. As for the screen-level temperature, sensitivity results are included, which will be addressed in Section 3.2.

  For the NU class, a negative bias of $-1.53\,\mathrm{K}$ is found for the day-time LST samples and a positive bias of $4.36\,\mathrm{K}$ for the
10 night-time samples for both REF and STD. The too low diurnal cycle is consistent with the results for the screen-level air temperatures (see Section 3.1.1). The model results for REF confirm the observed gradual increase of LST with the increasing urban density. At the same time, the day- and night-time SUHI increases as well with the urban density in both REF and the





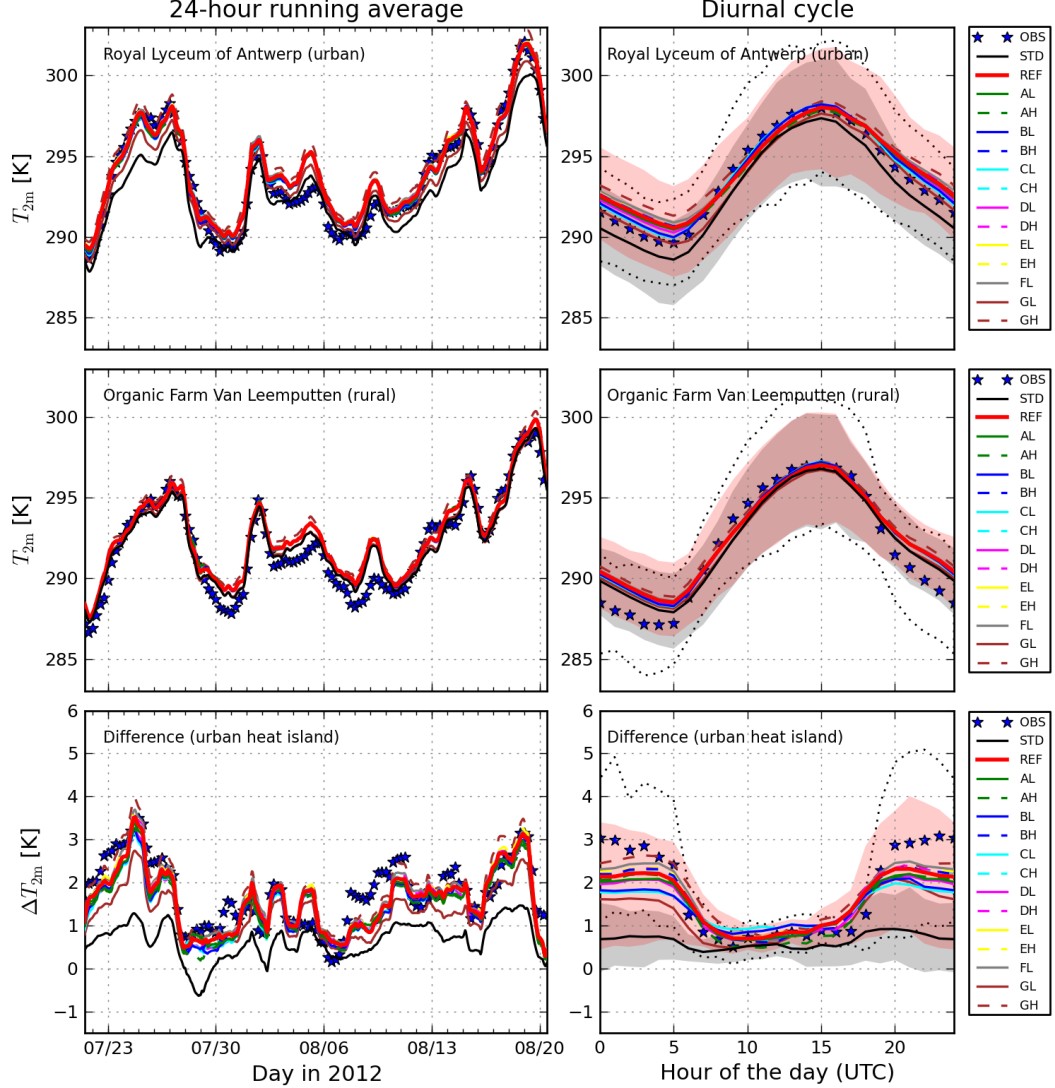

**Figure 3.** 24-hour running averages and mean diurnal cycle for modelled (thick lines) and observed (OBS; blue stars) temperatures during mid-summer (2012/07/21 until 2012/08/20) at the Royal Lyceum for Antwerp (Urban), the Organic Farm Van Leemputten (Rural), and their difference (the Urban heat island effect of Antwerp). The simulation with the COSMO-CLM model coupled to the standard land-surface module TERRA_ML without urban parametrization is indicated with STD (black), whereas the reference simulation with the COSMO-CLM model + TERRA_URB version 2.0 with urban parametrization is indicated with REF (red). The dotted lines indicate the range between the 16th and 84th percentile of the observed temperatures, whereas the grey and red areas indicate the ranges for the simulations STD and REF, respectively. An overview of the canopy parameter sensitivity simulations (AL, AH, BL, BH...) can be found in Table 2.





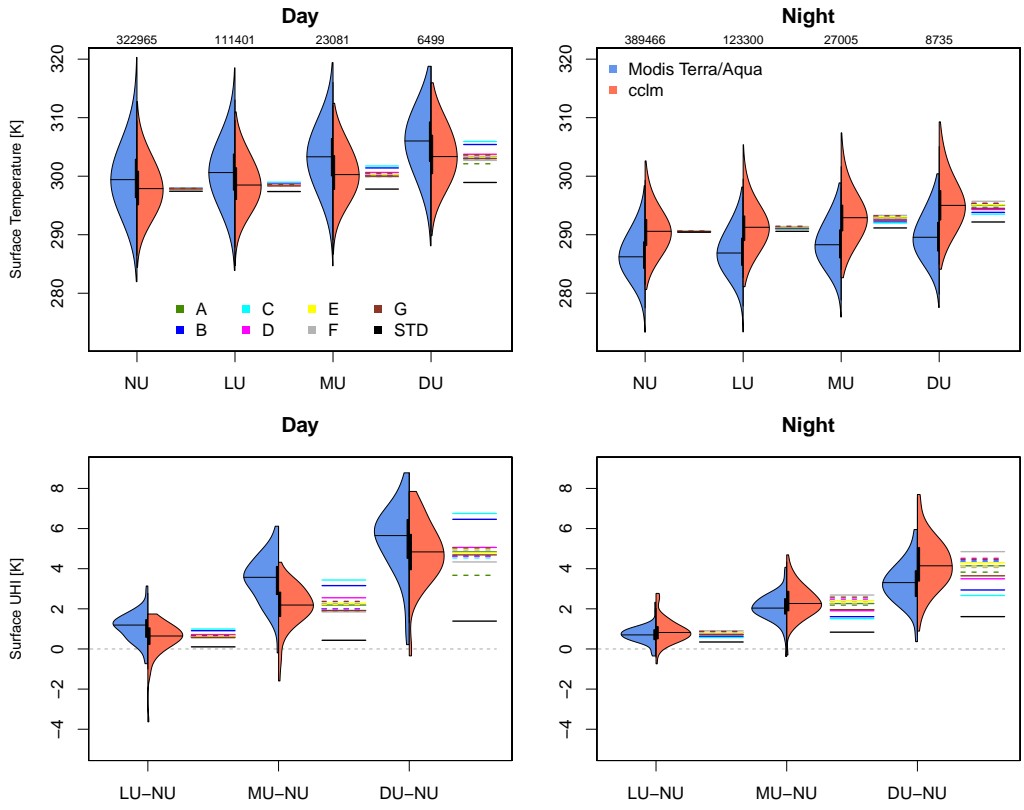

**Figure 4.** Evaluation of modelled land-surface temperatures against MODIS satellite observations. A distinction is made between four urban classes: no urban (NU) with Impervious Surface Area (ISA) $\leq 0.05$; light urban (LU) , $0.05 < ISA \leq 0.25$); medium urban (MU), $0.25 < ISA \leq 0.5$; dense urban (DU), $ISA > 0.5$. The amount of valid pixels are shown above each day and night plot. The violin plots represent the distribution of LST for MODIS in blue and the reference simulation (REF) with COSMO-CLM coupled to TERRA_URB version 2.0 in orange. Full and dashed lines represent the median for the low (AL, BL, ..., GL) and high (AH, BH, ..., GH) scenarios from Table 2, respectively. The upper panels show the sample distributions of the absolute temperatures, whereas the lower panels show the sample distribution of the difference between median of each urban class (LU, MU and DU) with the NU class.




observations. For the day, the modelled SUHI by REF has a negative median bias of $-1.52\,\text{K}$ and $-1.14\,\text{K}$ for the MU and DU class respectively, whereas it is $-0.6\,\text{K}$ for the LU class. For the night, the averaged modelled SUHI by REF matches very well the observations. Hereby, the LU, MU and DU classes are $0.03\,\text{K}$, $0.26\,\text{K}$ and $1.09\,\text{K}$. This general positive nocturnal bias is in contrast to the negative bias for the screen-level air temperatures (see Section 3.1.1). In contrast to REF, STD is not able
to capture the SUHI.

### 3.1.3 Nocturnal boundary layer

The model results are evaluated with observations of the nocturnal boundary-layer described in Section 2.4.3. Besides absolute temperature profiles, we also focus on the performance of the nocturnal BLUHI. The latter is calculated from the difference between temperature profiles in an industrial and natural area. The results are shown in Figure 5 and Table 6. As before,
sensitivity results are included, which will be addressed in Section 3.2. Because of their height, it needs to be noted that profile differences between the two mast towers also originate from other sources that are not related to the urbanization. In particular for certain wind directions, substantial influence could arise from the Scheldt river nearby the tower of Zwijndrecht.

The REF simulation is capable of capturing the temperature profiles and the variability of both towers. For the industrial site, the model profile matches well that of the observations with a positive bias of $0.41\,\text{K}$. A higher positive temperature
bias is found for the rural site ($0.73\,\text{K}$), which generally stems from the temperature overestimation near the ground. Hereby, the increasing temperature with height ($\simeq$ stable profile) in the model output is overestimated. These results of overestimated temperatures near the ground are consistent with the positive nocturnal bias in the screen-level measurements shown in Section 3.1.1. The model results for REF confirm the tendency towards destabilised temperature slopes at the industrial site in contrast to the more stable temperature slope at the rural site. As a result, the observed boundary-layer UHI effect ($1.02\,\text{K}$), calculated as
the difference between the temperature profile between the industrial and the natural site, is well reproduced by REF ($0.70\,\text{K}$) with a negative bias of $-0.32\,\text{K}$ and a correlation of $0.58$.

The positive bias for STD ($0.31\,\text{K}$) for the rural tower is slightly lower than that of REF. This is explained by the advection of urban heat towards the rural site in REF which is not present in STD. A much larger temperature shift between STD and REF occurs for the industrial tower (ZWN). The boundary-layer temperatures in STD are underestimated by $-0.86\,\text{K}$, which
is in contrast to REF for which it is overestimated. As the tendency towards more neutral/unstable temperature slopes in the industrial site is absent in STD, the boundary-layer UHI could not be reproduced with a negative bias of $-1.17\,\text{K}$ and a correlation $0.29$.

### 3.2 Urban parameter sensitivity

The model sensitivity experiment in response to the different urban input parameters is performed by means of model-bias
analysis. Hereby, the model output and performance statistics in terms of the observed temperature quantities (described in 2.4) for the different experiments (AL, AH, ..., GL, GH) described in Section 2.3 are compared with the reference simulation (REF) and the simulation without urban parametrization (STD). The results of the sensitivity experiments can be found in Figures 3, 4 and 5, and in Tables 3, 4, 5 and 6 (Taylor diagrams are not shown). In general, the different model sensitivities are





| VMM Zwijndrecht (industrial) | | | |
|---|---|---|---|
| EXP-ID | L | H | \|H - L\| |
| OBS | 291.58 ( - ) | - | - |
| STD/REF | 290.72 (-0.86) | 291.99 (+0.41) | 1.27 |
| A | 291.99 (+0.41) | 291.94 (+0.36) | 0.05 |
| B | 291.69 (+0.11) | 292.04 (+0.46) | 0.35 |
| C | 291.62 (+0.04) | 292.06 (+0.48) | 0.44 |
| D | 291.82 (+0.24) | 292.07 (+0.49) | 0.25 |
| E | 291.89 (+0.31) | 292.13 (+0.55) | 0.24 |
| F | 292.18 (+0.60) | 291.99 (+0.41) | 0.19 |
| G | 291.35 (-0.23) | 292.48 (+0.90) | 1.13 |
| SCK/CEN Mol (rural) | | | |
| EXP-ID | L | H | \|H - L\| |
| OBS | 290.56 ( - ) | - | - |
| STD/REF | 290.87 (+0.31) | 291.29 (+0.73) | 0.42 |
| A | 291.29 (+0.73) | 291.34 (+0.78) | 0.05 |
| B | 291.24 (+0.68) | 291.33 (+0.77) | 0.09 |
| C | 291.19 (+0.63) | 291.34 (+0.78) | 0.15 |
| D | 291.21 (+0.65) | 291.35 (+0.79) | 0.14 |
| E | 291.27 (+0.71) | 291.36 (+0.80) | 0.09 |
| F | 291.37 (+0.81) | 291.33 (+0.77) | 0.04 |
| G | 291.13 (+0.57) | 291.58 (+1.02) | 0.45 |
| Difference (urban heat island) | | | |
| EXP-ID | L | H | \|H - L\| |
| OBS | 1.02 ( - ) | - | - |
| STD/REF | -0.15 (-1.17) | 0.70 (-0.32) | 0.85 |
| A | 0.70 (-0.32) | 0.59 (-0.43) | 0.11 |
| B | 0.45 (-0.57) | 0.71 (-0.31) | 0.26 |
| C | 0.44 (-0.58) | 0.72 (-0.30) | 0.28 |
| D | 0.61 (-0.41) | 0.72 (-0.30) | 0.11 |
| E | 0.61 (-0.41) | 0.77 (-0.25) | 0.16 |
| F | 0.82 (-0.20) | 0.66 (-0.36) | 0.16 |
| G | 0.23 (-0.79) | 0.89 (-0.13) | 0.66 |

**Table 6.** Modelled and observed averages for the nocturnal profile temperatures (in K) for the towers on the VMM industrial site in Zwijndrecht and on the SCK/CEN rural site in Mol, during mid-summer (2012/07/21 until 2012/08/20). Each first row shows results of the observations. Each second row shows results for COSMO-CLM without urban parametrization (L column), for the COSMO-CLM using TERRA_URB version 2.0 (H column) and their absolute difference (R column). The remainder rows show each of the urban-parameter sensitivity simulations in Table 2. Hereby, the Low scenarios are shown in the L column, the High scenarios in the H column and the parameter sensitivity Range (absolute difference between L and H) in the $|H - L|$ column.





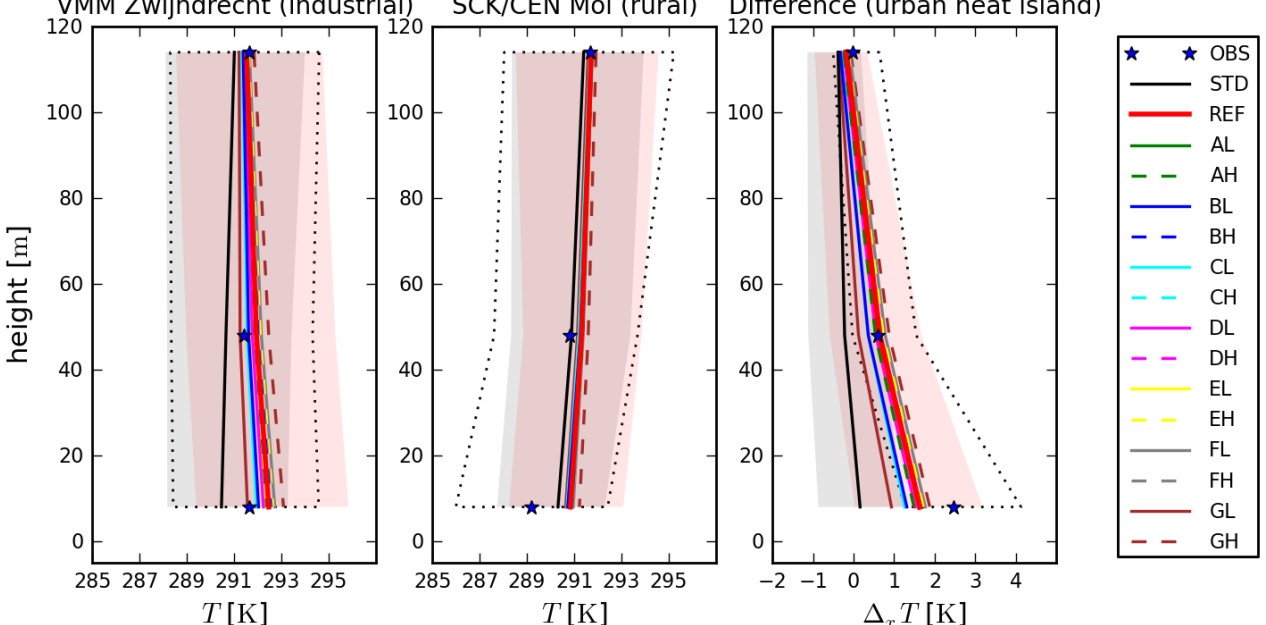

**Figure 5.** Observed (stars) and modelled (lines) nocturnal (0H) vertical profiles for VMM industrial site in Zwijndrecht and the SCK/CEN rural site in Mol, averaged for the summer period 2012/07/21 until 2012/08/20. STD (black) indicates the simulation with standard model COSMO-CLM without urban parametrization. REF uses the same setup as STD, but showing results of TERRA_URB version 2.0 coupled to COSMO-CLM. An overview of the remaining simulations (AL, AH, BL, BH...) can be found in Table 2. The dotted lines indicate the range between the 16 and 84th percentile of the observed profiles, whereas the grey, red and red areas indicate the ranges for the simulations STD and REF, respectively.

restricted to the urban areas, and therefore they emerge as changes in the UHIs. The model sensitivity is much smaller than the model changes when including urban parametrization (REF minus STD). This is the case for screen-level temperatures, LST and nocturnal boundary-layer temperatures, except for the change in the screen-level and boundary-layer temperatures when altering AHE.

5     With respect to the screen-level temperature and CLUHI intensity for Antwerp, the sensitivity range is generally larger during the evening and during the night when the CLUHI remains close to its maximum. The largest sensitivity range for the diurnal cycle is found for the parameters AHE and (to a smaller extent) the thermal parameters, which are the substrates' heat conductivity and heat capacity. A medium sensitivity range to the diurnal cycle is found for the building height, $\frac{H}{W}$-ratio and the roof fraction. The overall lowest model sensitivity is found from the substrate albedo, yet at day-time, when overall
10   sensitivity changes are the lowest, it is higher than the sensitivity from the thermal parameters. For the daily averaged CLUHI,



the magnitude of the different sensitivities are comparable. A more detailed description of the different parameter sensitivities is found below:

- An increased AHE obviously leads to an increase in screen-level urban temperatures and heat island intensity. Even though the AHE is higher during the day, the sensitivity range from AHE is largest during the night because of the confinement in the nocturnal boundary layer (Wouters et al., 2013; Bohnenstengel et al., 2011).

- A diurnal change in sign of the sensitivity is found for the thermal parameters: Lower thermal parameter values lead to a temperature increase during the day followed by a decrease during the night. This is due to the fact that a smaller portion of excess heat is needed to heat up the urban canopy during the day, hence leads to higher day-time temperatures. At the same time, less excess heat buffering in the urban canopy occurs which lowers the temperatures and night-time. The sign change results in a relatively low sensitivity of the daily-mean temperature, even though the day- and night-time sensitivities are relatively high.

- An overall decrease in day- and night-time CLUHI appears from the decrease in building height. The consequential lower roughness length (Equation 15) results in lower wind speed. As a result, the urban excess heat is less accumulated in the urban centres lowering the air temperatures there. Smaller buildings also yield a lower effective heat capacity and heat conductivity of the soil layers below (see Equations 5 and 7). In the same way as in the previous point concerning the thermal parameters, such a change counteracts (yet not fully) the above-mentioned day-time temperature decrease and enhances the night-time temperature decrease.

- Both the increase in roof fraction and substrate albedo lead to an overall decrease in nocturnal and day-time UHI, because they both increase effective albedo according to Equation 9. It is remarkable that the sensitivity to the roof fraction is larger than that for the substrate albedo at night-time, whereas this becomes reversed at day-time. This means that another process different from the effective albedo effect is also important: In fact, a larger roof fraction also implies a reduction in effective heat capacity and heat conductivity (see Equations 3, 4 and 6). Again in the same way as in the previous point concerning the thermal parameters, this counteracts the decrease in day-time UHI, while it enhances the decrease in night-time UHI.

- A lower $\frac{H}{W}$-ratio leads to an overall decrease in screen-level urban temperatures and UHI for the day and for the night. Two mechanisms are at play here: on the one hand, a decrease in short-wave radiative trapping in the canyon (see Equations 9 and 10) leads to a higher effective albedo, hence a reduced conversion of solar radiation into heat. On the other hand, a lower $\frac{H}{W}$-ratio decreases the heat transfer below the surface due to the lower contact surface (see Equation 3) implying a lower effective heat capacity and conductivity (see Equations 4 and 6). Combined, they lead to an additional decrease in nocturnal urban temperature and UHI, while counteracting the day-time decrease resulting from the higher effective albedo mentioned above.

The sensitivity results for LST sensitivity are consistent with the sensitivity to the screen-level temperatures. For example, lower substrate heat conductivity or heat capacity leads to higher urban LST's and SUHI at day-time, and lower urban LST's





and SUHI at night-time. However, the hierarchy of sensitivity to LST and SUHI is different from that of the screen-level temperature: the largest sensitivity range for the urban LST and SUHI now relates to the thermal parameters and not to the change in AHE which now has a medium sensitivity range. This is explained by the fact that AHE is considered as a heat source term to the first model layer in the model, hence only indirectly influences the LST. Sensitivity of LST to the other

urban canopy parameters are similar to that of the screen-level temperatures: The $\frac{H}{W}$-ratio and roof fraction yield a medium sensitivity range, and the building height yields a low sensitivity range to LST. The sensitivity to the substrate albedo is low for the night-time LST and high for the day-time LST. The sensitivity results for the temperature profiles show a similar sensitivity as for the screen-level temperatures. Hereby, the model sensitivity propagates throughout the nocturnal boundary layer for at least the first 110 m above ground level, hence it modulates the BLUHI.

Analysis of the model biases and Taylor-plots for each of the temperature quantities (not shown) demonstrate that the model performance change for each of the parameter sensitivity runs is ambiguous, and depends on the temperature quantity considered. On the one hand, parameter changes leading to a better performance in CLUHI and BLUHI sometimes lead to worse performance in terms of absolute temperatures. In particular, a decrease in the value of the thermal parameters leads to a lower positive model bias and better model performance in terms of absolute screen-level temperatures, but a larger negative

bias and worse model performance in terms of CLUHI. On the other hand, the performance change between the SUHI and CLUHI may lead to a divergent behaviour as well: an overall improvement in the CLUHI (lower positive day-time bias; lower negative night-time bias) by increasing the value of the thermal parameters leads to an overall deterioration of the SUHI (higher day-time negative bias; higher night-time positive bias). Finally, the day- and night-time performance changes are also divergent. For example, increasing the surface albedo leads to an overall deteriorated performance in day-time LST and SUHI,

but leads to an improved night-time performance.

## 4 Discussion and conclusions

The Semi-empirical URban-canopY parametrization SURY is developed for efficiently representing urban physics in atmospheric models. The methodology bridges the gap between bulk and explicit-canyon schemes in atmospheric models. Hereby, the urban canopy parameters are translated into bulk parameters. The latter can be easily taken into account in bulk urban

land-surface schemes in existing atmospheric models. The urban canopy parameters as input for SURY include the canyon height-to-width ratio, the building height, the roof fraction, and the short-wave albedo, thermal emissivity, heat conductivity and heat capacity of the surface substrate. The bulk parameters as output of SURY include the bulk parameters for albedo, emissivity, aerodynamic and thermal roughness length, the ground heat conductivity and heat capacity (or alternatively the thermal transmittance). The methodology delivers theoretical and empirically-verified robustness that is based on detailed ur-

ban observational and modelling experiments. Additional model robustness has been provided by comparing existing bulk parameters from top-down estimates with those translated from bottom-up urban canopy parameter inventories by the urban canopy parametrization.



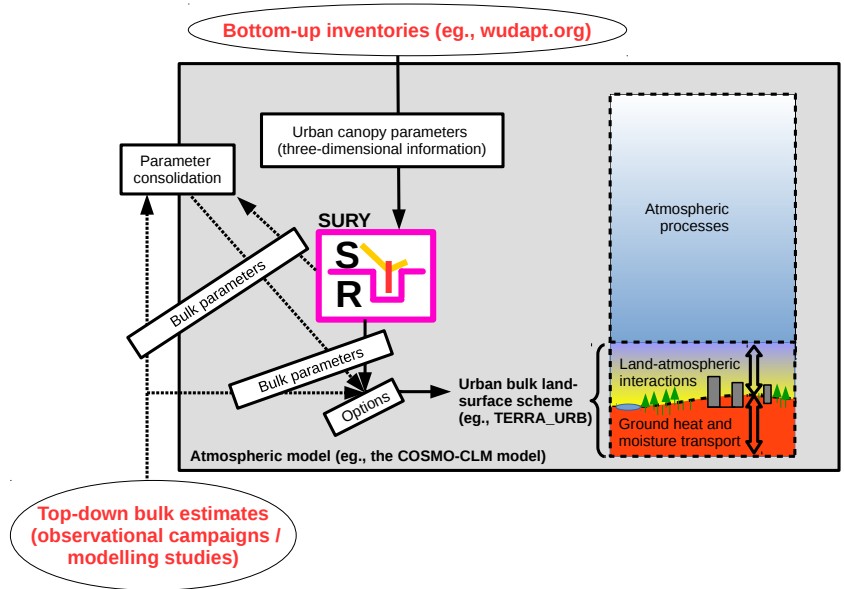

**Figure 6.** Outline of the Semi-empirical URban canopY parametrization SURY implemented in an atmospheric model system. The main application flow of SURY is indicated with full arrows, whereas optional application flows are indicated with dashed arrows.

The outline of SURY implemented in an atmospheric model system is given in Figure 6. SURY in combination with a bulk urban-land surface schemes provides several benefits over other methodologies for urban canopy parametrization with regard to atmospheric modelling. They are related to either applicability, versatility, computational cost and model consistency:

– The framework allows for consistency in switching between bulk parameters from top-down estimates and urban canopy
parameters from detailed bottom-up inventories, hence enhances its applicability and versatility: On the one hand, SURY allows to employ detailed urban canopy information as soon as they are available. On the other hand, it can still fall back on bulk parameters when more detailed urban canopy parameters are missing, hence avoiding deterioration of the model performance. SURY further allows for consolidating top-down bulk parameter datasets and those translated from urban canopy parameter datasets. It also facilitates modelling consistency between areas having urban canopy parameter
datasets and other areas having bulk parameter datasets. Finally, the framework enables the analysis of the propagation of uncertainty in the input parameters by comparing simulations using bulk parameter estimates with those using urban canopy parameter datasets.

– SURY combined with a bulk urban land-surface parametrization is beneficial in terms of computational cost in atmospheric modelling applications. In particular, the bulk albedo approximation avoids explicit numerical computation of
the complex canyon radiation trapping, hence largely reduces the computational demand compared to other explicit canyon-radiation models. In case of the COSMO(-CLM) model, one only measures a computational overhead of 7%




over the original COSMO(-CLM) model without urban land-surface parametrization. The majority of the computational overhead stems from the implementation of the tile approach in TERRA_URB and the overhead from SURY itself is negligible.

- As SURY translates urban canopy parameters into bulk parameters, it can be easily applied for intrinsic implementation of urban physical processes and their dependency on urban canopy parameters in existing atmospheric models employing a bulk urban land-surface scsheme. An intrinsic implementation instead of an external urban land-surface parametrization preserves consistency between the representation of the different physical process and features in urban environments on the one hand and that of natural environments on the other hand. Hereby, it is assured that the contrasts in model response between urban areas and natural areas only stems from the differential land-surface surface parameter values. In this way,

artificial discrepancies that arise from possible different formulations between an external land-surface module resolving the urban areas and an internal land-surface module resolving the natural areas are avoided. Furthermore, it allows for the transparent application of intrinsic features of the host atmospheric model in the urban-physics modelling, such as the representation of vegetation (shading) in the urban canopy. In the case of the COSMO(-CLM) model, these features include the multi-layer snow representation and the TKE-based surface-layer turbulent transfer scheme.

SURY is evaluated in online mode with the COSMO(-CLM) model. Therefore, the COSMO(-CLM) model is extended with the bulk urban land-surface scheme TERRA_URB version 2, which allows for taking urban bulk parameters from SURY into account. An online model evaluation over the Belgian urban extent during summer 2012 demonstrates that the model is capable of capturing urban climate characteristics. The model captures well the daily and diurnal variability of the UHI in terms of land-surface temperatures, screen-level temperatures and nocturnal boundary-layer temperatures. Hereby, most of the

negative temperature biases occurring for the urban areas in the original COSMO(-CLM) model without urban parametrization are alleviated.

The model sensitivity is investigated for which SURY takes urban canopy parameter ranges of the Local Climate Zones of 'compact low-rise' and 'compact mid-rise' in Stewart and Oke (2012). The model response and performance change to urban canopy parameter changes are generally restricted to the urban areas. They are also smaller than the effect of the introduc-

tion of the urban land-surface scheme in the atmospheric model. With regard to the screen-level temperatures, the nocturnal boundary-layers temperature and associated CLUHI and BLUHI, the largest model sensitivity is found for the anthropogenic heat emission (AHE) and the urban thermal parameters, a medium sensitivity for the building hight and $\frac{H}{W}$-ratio, and the lowest sensitivity for roof fraction. With regard to the day- and night-time LST and SUHI, the largest model sensitivity is found for the urban thermal parameters, a low sensitivity for the AHE, $\frac{H}{W}$ and roof fraction, and a very low sensitivity for the building height.

For both SUHI and CLUHI, the sensitivity to the albedo is relatively high yet slightly lower than for the thermal parameters during day-time whereas it is slightly lower than AHE, $\frac{H}{W}$ and roof fracton during night-time.

The following recommendations are made with regard to future urban-climate research:

- The city-scale effect of the urban-parameter changes on modelling urban heat islands demonstrates that uncertainty and variability of urban parameters needs to be taken into account in regional climate modelling and numerical weather





prediction for cities. It also demonstrates that supplementary urban information has potential for improving urban regional climate modelling. Hereby, the priority should be given to those parameters that are causing the largest sensitivity listed above. The latter are the substrate thermal parameters and the anthropogenic heat emissions. Note that these are the parameters that nowadays depend on detailed inventories, which are generally much more difficult to acquire than the radiative and morphological parameters. It demonstrates the necessity of improving existing and developing new methodologies for more automatized acquisition of these parameters.

– The model performance and its sensitivity to the different canopy parameters largely depend on the temperature quantity considered. More particular, parameter settings leading to better UHI sometimes lead to worse absolute temperatures and vice versa, which is also the case for day-time temperatures versus night-time temperatures, and land-surface temperatures versus air temperatures. This ambiguity demonstrates that a multi-variable model evaluation is a requirement for improving and comparing urban-climate modelling strategies.

– Although TERRA_URB version 2.0 implementing SURY offers a general model improvement with regard to urban temperatures and urban heat islands, it could not alleviate other systematic errors in the COSMO(-CLM) model. This is especially the case for the rural absolute temperatures for which the urban effect is small. One should keep in mind that systematic errors could be propagated, amplified or compensated by the urban-atmospheric interactions, hence deteriorating urban-climate modelling (assessment). For example, an underestimation of the nocturnal air-temperature heat islands may originate from the overestimation of nocturnal temperatures found in rural areas. Moreover, some of these errors overrule the model uncertainty range with regard to the urban canopy parameters. This demonstrates that the majority of the model uncertainty is not related to urban canopy parameter uncertainty, but from deficiencies in the land-surface module and other aspects of the coupled atmospheric system. Such systematic errors may also (partly) cause the ambiguity in model performance and its sensitivity to the urban parameters. In addition to the improvement of urban land-surface parametrization and datasets, urban-climate modelling could largely benefit from improving other components of the coupled atmospheric model system.

– Besides the advantages of SURY combined with bulk land-surface schemes listed above, its semi-explicit natre also implies some limitations with respect to complex urban physical processes in an urban environment. The heterogeneity of the urban canopy induces micro-scale dynamic and physical processes in the urban canopy, such as shadowing and buoyancy flows, and complex inner-building heat and moisture transfer. As a result, induced micro-scale features, such as heterogeneous temperatures patterns and wind gusts, are not explicitly resolved. In order to explicitly resolve such features, one requires detailed urban-physics modelling. They include the more explicit radiation schemes allowing heterogeneous temperatures between sun-lit and shaded facets (eg., Schubert et al., 2012), computational fluid dynamics (eg., Allegrini et al., 2014), and inner-building energy models (eg., Bueno et al., 2012; Crawley et al., 2000). Yet, these methodologies obviously require much finer meshes, larger computational resources and very detailed information on urban design. With regard to the practical implementation of numerical weather prediction and long-term regional climate modelling at the regional scales nowadays, the advantages of SURY outweigh the limitations. In view of providing un-





certainty spread with regard to urban climate-change projection, we also recommend SURY for intrinsic implementation in other existing atmospheric numerical models. In case there's need for more detailed urban-climate impact assessment, urban land-surface parametrizations with higher complexity need to be considered.

**Code availability**

SURY is provided as a documented Python module freely available with this paper. It should work on any platform with Python and NumPy installed. The presented model version for SURY is 1.0. All comments, questions, suggestions and critiques regarding the functioning of the Python routine can be directed to the author of this paper. The modified version of the COSMO-CLM model used in this study with TERRA_URB that implements SURY is accessible on the CLM-community website with a member login.

**Appendix A:  TERRA_URB version 2.0**

This appendix describes the second version of TERRA_URB, the bulk urban land-surface scheme of the COSMO(-CLM) model. Modifications to the surface-layer turbulent transport scheme of the COSMO(-CLM) model, the modified surface evaporation and transpiration, the anthropogenic heat emission, the tile approach and the additional surface input parameters are successively given in the subsections below. Bulk parameters are calculated from the urban canopy parameters with the

Semi-Empirical URban CanopY parametrization (SURY) described in Section 2.1, see also Table 1.

**A1   Surface-layer turbulent transport**

In contrast to its previous version employing the similarity-based turbulent transfer scheme of Monin and Obukhov (1954), TERRA_URB version 2.0 makes use of the TKE-based surface-layer turbulent transfer scheme of the COSMO(-CLM) model. Herein, the exchanges for momentum and heat within the surface layer between the surface and the lowest atmospheric model

layer are determined as follows:

$$(\overline{u'w'})_0 \quad = \quad -u_*^2 = -u_{\mathrm{A}}/r_{\mathrm{SA}}^M, \tag{A1}$$

$$Q_{\mathrm{H}} = \rho c_p (\overline{w'\theta'})_0 \quad = \quad -\rho c_p u_* \theta_* = -\rho c_p (\theta_{\mathrm{A}} - \theta_{\mathrm{s}})/r_{\mathrm{SA}}^H, \tag{A2}$$

where $\rho$ and $c_p$ are the density and the specific heat of the air, $u_{\mathrm{A}}$ and $\theta_{\mathrm{A}}$ are the absolute wind speed and potential temperature at the lowest atmospheric model layer, $\theta_{\mathrm{S}}$ is the potential temperature of the surface, $u_*$ is the friction velocity, and $\theta_*$ is the

turbulent temperature scale. Stability-dependent transfer-layer resistances (ie., resistances to the surface-layer turbulent transfer between the Surface and the lowest Atmospheric model layer) for momentum ($r_{\mathrm{SA}}^M$) and heat ($r_{\mathrm{SA}}^H$) are calculated with the TKE-based surface-layer transfer scheme of the COSMO(-CLM) model (Doms et al., 2011). For the urban canopy, aerodynamic roughness length is obtained from Equation 15. For the natural land cover, $z_0$ is adopted from standard input parameters of the COSMO(-CLM) model, see Section 2.2. The thermal roughness length $z_{0\mathrm{H}}$ is obtained with a parametrization of the inverse



Stanton number (as in De Ridder, 2006; Demuzere et al., 2008):

$$kB^{-1} = \ln\left(\frac{z_0}{z_{0\mathrm{H}}}\right) \tag{A3}$$

with $k$ the von Kàrmàn constant. For the natural areas, the inverse Stanton number is diagnosed from the TKE-based surface-layer transfer scheme as follows:

$$kB^{-1} = r_{\mathrm{S0}}^{H}\sqrt{2e}S_{\mathrm{H}}(z_0) \tag{A4}$$

where $r_{\mathrm{S0}}^{H}$ is the roughness-layer resistance for heat (ie., the resistance to the surface-layer turbulent heat transfer between the surface and $z_0$), $e = 0.5(u^{'2} + v^{'2} + w^{'2})$ the turbulent kinetic energy, and $S_H(z_0)$ the stability function for the Prandtl layer. For the urban canopy, the bluff-body thermal roughness length is used from Equation 17 Inserting Equation 17 into Equation A4, it leads to a modified $r_{\mathrm{S0}}^{H}$. In turn, this enters the surface-layer heat transfer equation (Equation A2) as follows:

$$r_{\mathrm{SA}}^{H} = r_{\mathrm{S0}}^{H} + r_{\mathrm{0A}}^{H} \tag{A5}$$

where $r_{\mathrm{0A}}^{H}$ is the free atmospheric resistance (ie., the resistance between the surface-layer turbulent transfer $z_0$ and the lowest atmospheric model layer).

## A2    Evaporation and transpiration

In contrast to its previous version only considering the water on the impervious surfaces, TERRA_URB version 2.0 is also capable of explicitly representing the water-permeable surfaces (bare soil), vegetation and snow in the urban canopy. The evapotranspiration is obtained in a similar way as for the natural land in the standard soil module TERRA_ML (see pp. 111 of Doms et al., 2011) as follows:

$$E = T + f_b(E_b + E_i) + f_{\mathrm{imp}}E_{\mathrm{imp}} + E_{\mathrm{snow}} \tag{A6}$$

where $T$ the transpiration from vegetation, $E_b$ and $E_i$ are respectively the evaporation rates from the (water-permeable) bare-soil fraction $f_b$ and that from the intercepted water on the bare soil and vegetation, and $E_{\mathrm{snow}}$ the snow evaporation. In addition to the standard surface module TERRA_ML, the evaporation from the water storage $E_{\mathrm{imp}}$ on the impervious surface fraction $f_{\mathrm{imp}} = 1 - f_b$ is taken into account. Herein, $E_b$ is calculated as follows:

$$E_b \;\; = \;\; (1 - f_{\mathrm{snow}})\min(E_p, F_m)H(E_p), \tag{A7}$$

$$\tag{A8}$$

where $E_p$ is the potential evaporation (when positive) or condensation (when negative), $H$ is the Heaviside function which yields 1 in case of evaporation ($E_p \geq 0$) and 0 in case of condensation ($E_p < 0$), and $F_m$ the maximum moisture flux the soil can sustain. The potential evaporation $E_p$ is obtained from:

$$E_p = -\rho_w(q_v - q_{sat}(T_s))/r_{\mathrm{SA}}^{H} \tag{A9}$$





where $r_{\mathrm{SA}}^{H}$ is the surface-layer transfer resistance for heat (or scalars), $q_v$ and $u_a$ are respectively the specific humidity and absolute wind speed at the half level of the first model layer, and $q_{sat}(T_s)$ is the saturated specific humidity at the surface temperature $T_s$. The calculation of $E_{\mathrm{imp}}$ is covered in the next appendix.

## A3 Impervious water storage

The evaporation from the water storage $E_{\mathrm{imp}}$ on the impervious surface fraction is calculated as follows:

$$E_{\mathrm{imp}} = (1 - f_{\mathrm{snow}})E_p H(E_p)\,\delta_{\mathrm{imp}} \tag{A10}$$

flux through the surface that the soil can sustain (Dickinson, 1984; Doms et al., 2011).

The evaporative area fraction $\delta_{\mathrm{imp}}$ of the impervious area fraction $f_{\mathrm{imp}}$ is calculated according Wouters et al. (2015):

$$\delta_{\mathrm{imp}} = \delta_m \left(\frac{w}{w_m}\right)^{2/3} \tag{A11}$$

where $\delta_m$ is the maximum evaporative surface fraction and $w_m$ the maximum water storage capacity on the impervious surface. Eqn. A11 was obtained by assuming a probability density function of water reservoirs for which the density constantly decreases with increasing reservoir depth.

The amount of water $w$ on the impervious surface changes as follows:

$$\frac{dw}{dt} = (1 - f_{\mathrm{snow}})(-E_{\mathrm{imp}} + C + P_r) - R_{\mathrm{imp,runoff}} - R_{\mathrm{imp,infil}} \tag{A12}$$

where $P_r$ is the rain rate, $C = -E_p H(-E_p)$ is the condensation, $R_{\mathrm{imp,runoff}}$ is the runoff to sewerage and rivers, and $R_{\mathrm{imp,infil}}$ is the infiltration rate of water into the natural soil originating from the impervious surface (for example, by means of infiltration tubes). The runoff and/or soil infiltration of water hitting the impervious surface occurs as soon as the maximum water storage $w_m$ is exceeded:

$$R_{\mathrm{imp,runoff}} = (1 - f_{\mathrm{snow}})H(w - w_m)(C + P_r)c_{\mathrm{imp,runoff}} \tag{A13}$$

$$R_{\mathrm{imp,infil}} = (1 - f_{\mathrm{snow}})H(w - w_m)(C + P_r)(1 - c_{\mathrm{imp,runoff}}) \tag{A14}$$

where $c_{\mathrm{imp,runoff}}$ is defined as the runoff index of the impervious surface, for which the value depends on the presence of infiltration systems: A value equal to one means that all rainwater exceeding the maximum water-storage threshold from streets and roofs $w_m$ is directed to the sewerage and rivers, whereas values equal to zero means that it infiltrates into the ground.

Default values for $w_m = 1.31\,\mathrm{kg\,m^{-2}}$ and $\delta_m = 0.12$ are taken from Wouters et al. (2015). By default, all water overshoot is considered as runoff, hence $c_{\mathrm{imp,runoff}}$ is set to one by default, meaning that all rainwater overshoot from streets and roofs is considered to be directed to sewerage and rivers.

## A4 Anthropogenic heat emission

TERRA_URB accounts for Anthropogenic Heat Emission (AHE) from human activity, which includes the energy dissipation from combustion and electricity consumption. It originates from heating and cooling (such as air conditioning) of buildings, and



traffic, but also from domestic, industrial and agricultural activity (Sailor, 2011). The AHE is calculated following the methodology of Flanner (2009), see also http://www.cgd.ucar.edu/tss/ahf/. It takes into account latitudinally-dependent seasonal and diurnal distribution functions (Flanner, 2009, their Figures 1 and 2) that are superimposed on annual-mean Anthropogenic Heat Flux (AHF). The AHE is added to the surface sensible heat flux, hence acts as an additional heat source to the surface layer.

## A5 Tile approach

TERRA_URB makes a distinction between the urban canopy and natural land-cover. This is done for each grid-cell with a tile approach. Herein TERRA_URB is called twice, once for the urban canopy and once for the natural land cover specifying a different set of bulk parameters. On the one hand, the surface input parameters for the urban canopy are obtained from urban canopy parameters with SURY. On the other hand, surface input parameters for the natural land-cover are provided by standard input parameters of the COSMO(-CLM) model. In this way, the ground heat and moisture transport and land-atmosphere exchanges in terms turbulent transport of momentum, heat and moisture are determined separately for each tile. The coupling to the atmospheric model is achieved by weighting each of the land-atmosphere fluxes according to the fractions of the urban canopy and natural land cover. The radiation exchanges are determined by grid-cell averaged value of albedo and emissivity, which is weighted according to the respective tile fraction.

## A6 Surface input parameters

In addition to the land-surface parameters for the standard COSMO(-CLM) model withouth urban land-surface parametrization described in Section 2.2, the implementation of TERRA_URB requires the two additional fields. They include the total Impervious Surface Area (ISA) and annual-mean Anthropogenic Heat Flux (AHF).

The ISA-field is obtained from the European soil sealing dataset representative for the year 2006 of the European Environmental Agency at $100\,\mathrm{m}$ resolution (Maucha et al., 2010). Optionally for regions outside Europe, the NOAA dataset (Elvidge et al., 2007) globally available at $1\,\mathrm{km}$ resolution can be selected as well. The latter offers flexibility for the extension with upcoming improved global products. In the default configuration of TERRA_URB version 2.0, $f_{\mathrm{imp}}$ equals to one for the urban canopy tile (all impervious) and zero for the natural land-cover tile (none impervious). As a result, the fraction of urban canopy at each grid-cell is equal to the value of ISA and the fraction of the natural land cover is its complement. All vegetation is resolved inside the natural land-cover tile, hence the vegetation abundance from the standard input parameters is increased for the natural land-cover tile according to its fraction, and set to zero for the urban canopy tile. All other parameters values specified for the natural land in TERRA_URB are in accordance to the original version of TERRA_ML and the COSMO(-CLM) model. As a result, the processes (regarding the ground-heat and water transport and land-atmosphere interactions) resolved for the natural land-cover have not been altered. The only exception is that the vertical profile formulation of the bulk heat capacity (Equation 5) and heat capacity (Equation 7) from SURY is also employed for the natural tile. Hereby, default values for $h$ and SAI for the natural land are equal to $0.01\,\mathrm{m}$ and 2, respectively. This was done for providing consistency between the urban canopy tile and the natural land-cover tile. Such a formulation also provides consistency with the TKE-based surface-layer turbulent transfer scheme in which SAI is also equal to 2.



The AHF-field is obtained from the global dataset of Flanner (2009), which is generated from country-specific data of energy consumption from non-renewable sources. This was apportioned according to population density (conserving the national total) and converted to annual-mean gridded energy flux at a resolution of $2.5 \times 2.5$ minutes. By default, a dataset is used for which these annual-mean values on the model grid are redistributed according to the ISA-field at a scale of $50\,\mathrm{km}$. Hereby, it is assumed that areas with large ISA fraction (including industrial areas with low population densities) have higher anthropogenic heat emissions. For Brussels, the Belgian Capital in the centre of the domain in Figure 1, the AHE reaches an annual-mean value of $49.16\,\mathrm{m^{-2}}$ in the city centre. These values are of the same magnitude to that obtained in Van Weverberg et al. (2008), ie. $43.6\,\mathrm{W\,m^{-2}}$. While not considering this as a formal validation, the similarity of magnitude of the results obtained by two very different methods inspires confidence. According to the latitudinally-dependent seasonal and diurnal distribution functions adopted from Flanner (2009), the values for summer (July) vary between $18.40\,\mathrm{W\,m^{-2}}$ (night-time) and $37.7\,\mathrm{W\,m^{-2}}$ (morning and afternoon). For winter (January), they vary between $41.1\,\mathrm{W\,m^{-2}}$ and $84.1\,\mathrm{W\,m^{-2}}$.

*Acknowledgements.* The work described in this paper has received funding from the Belgian Science Policy Office through its Science for a Sustainable Development Programme under Contracts SD/CS/041/MACCBET and BR/143/A2/CORDEX.be. It has also received funding from and from the Flemish regional government through a contract as an FWO (Fund for Scientific Research) post-doctoral position. The computational resources and services used in this work were also provided by the Hercules Foundation and the Flemish Government (department Economics, Sciences and Innovation - EWI). Additional assistance of resources provided at the NCI National Facility systems at the Australian National University through the National Computational Merit Allocation Scheme supported by the Australian Government. We would like to thank Gianluca Mussetti (Swiss Federal Laboratories for Materials Science and Technology - EMPA) and Johan Zürger (Austrian Institute of Technology - AIT) for testing TERRA_URB for regional-climate modelling over Zürich and Vienna. We also acknowledge Jüergen Helmert for providing support in extending the EXTPAR input parameter tool, and the COSMO consortium and CLM-community for support in the model setup.



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
