# Peer review of "The efficient urban-canopy dependency parametrization SURY (v1.0) for atmospheric modelling: description and application with the COSMO-CLM model (v5.0\_clm6) for a Belgian summer"

_Geoscientific Model Development, 2016_

## Referee Comment (RC1) · C. Meng (Referee) · 13 Apr 2016

General comments: This paper developed a semi-empirical urban parameterization scheme, and applied it to COSMO-CLM model; the validation and sensitivity analysis were also implemented. Overall, this paper is well organized and easily to read; but it still need to be improved and clarified in some key points, so I suggest this paper should be published after a MAJOR revision. 1. This paper boasts to present a Semi-empirical URban-canopY parametrization SURY, which bridges the gap between bulk urban land-surface schemes and explicit-canyon schemes. But it lacks the comparison

between the bulk urban schemes and explicit-canyon schemes. So, I question whether the SURY scheme is necessary or not. 2. As for SURY, why the author choses these parameters namely bulk albedo, bulk emissivity, etc. as the output of SURY, this need to be clarified. 3. Surface-Area Index (SAI) is a crucial important factor in this paper to reparametrize the ground heat transport parameters, but why SAI is chosen to do this? Why these parameters need to be reparametrized?

Specific comments: 1. Page 4 Table 1: I think these parameters should be reworked because they are varied with different areas. 2. Page 6 Equation 3: Please explain why use this equation to reparametrize the parameters. 3. Page 7 Equation 10 and 11: These equations also need to be explained. 4. Page 8 Line 22: In my opinion the z0 is the most important parameter in surface layer turbulent fluxes parametrization, so I think at least z0 should be added in the sensitivity analysis. 5. Page 29 Line 5: I think the author should provide a website of the models.
* * *

---

## Short Comment (SC1) · 13 Apr 2016

Dear Authors,

as Executive Editor of GMD I appreciate that you comply with the Editorial of GMD by including the model name and version number in the title of your article.

Nevertheless, already the first line of the abstract and the Code Availability Section show, that the development published here is SURY (v.1.0). You already took the effort to name it and apply a version number. Thus I very much favour to name SURY in the title as well.

[Figure]

I suggest to change the title in a way like "The efficient urban canopy parametrization for atmospheric modelling (SURY v1.0): description and application with the COSMO-CLM model (version 5.0_clm6) for a Belgian Summer"

or "The Semi-empirical URban-canopY parametrization SURY (v1.0): description and application with the COSMO-CLM model (version 5.0_clm6) for a Belgian Summer"

Best regards, Astrid Kerkweg
* * *

---

## Author Comment (AC1) · 2 May 2016

The authors would like to thank the referee for the review of the manuscript. We appreciate the suggestions for clarifying the manuscript in some of its key points. As part of the interactive discussion of GMDD, we provide a reply to the different reviewer's comments. A definitive answer will be provided at the time of the revision, in which the necessary changes will be indicated in the revised manuscript.

*General comments:*

*This paper boasts to present a Semiempirical URban-canopY parametrization SURY, which bridges the gap between bulk urban land-surface schemes and explicit-canyon schemes. But it lacks the comparison between the bulk urban schemes and explicit-canyon schemes....*

The authors agree that model intercomparison studies comparing bulk schemes on the one hand and the explicit-canyon schemes on the other hand are important to substantiate the development and advantages of SURY as an 'in-between' model approach. In this respect, *the introduction in our paper sumerizes a qualitative comparison between the bulk schemes and the explicit canyon schemes (see P2R9→P3R8). For the revised manuscript, we will add the references to studies that compare the different schemes as follows (at P2R10): "Even though their purpose of representing urban physics in land-surface schemes of atmospheric models is the same,* **intercomparison studies (Grimmond et al., 2011; Best and Grimmond, 2015; Trusilova et al.2015; Karsisto et al., 2015) demonstrate** *that they differ in terms of modelling strategy, complexity, input parameters and applicability* *."*

Furthermore, the intercomparison studies support our model development for bringing canopy-dependent urban physics to existing urban bulk land-surface schemes (see in bold just below), also allowing to consolidate urban canopy parameter datasets with urban bulk parameters (as stated in our conclusions). Therefore, the following information will be added to the revised introduction as well (at P2R29): "*At first, the urban canopy parameters, which include information about* **the threedimensional** *urban morphology and material properties, are obtained from detailed inventories (Loridan and Grimmond, 2012;Jackson et al., 2010).* **The first urban model intercomparison project demonstrate that such parameter information are important for improved modelling performance in existing urban land-surface schemes (Best and Grimmond, 2015).**"

*Finally, a full quantitative comparison between urban land-surface schemes, which is covered by existing studies as mentioned above, is outside of the scope of this paper about the specific development of SURY.*

*... So, I question whether the SURY scheme is necessary or not.*

The authors agree with the reviewer that the added value of SURY over existing methodologies needs to be stated very clear in the manuscript. Therefore, we will include the following information as bullet-points in the revised introduction (at P3R10):
- (*as already stated in the introduction*) Based on detailed observational studies, modelling experiments and available parameter inventories, SURY represents a robust translation of urban-canopy parameters containing three-dimensional information towards bulk parameters.
- the translation allows to combine advantages (hence bridges the gap) of both bulk schemes and explicit-canyon schemes in urban modelling studies. Especially, **it brings canopy-dependent urban physics** (used to be reserved for the explicit-canyon schemes before) **to the existing urban bulk land-surface schemes.** This could be done while preserving the low computational cost and low complexity of the bulk schemes.
- the translation offers versatility and consistency in choosing between urban-canopy parameters from bottom-up inventories and bulk parameters from top-down inventories.

Note that a more extensive discussion about advantages (and limitations) of SURY with regard to applicability, versatility, model consistency and the computational cost is provided in the 'discussion and conclussions'-section, see R26R1→R27R14 and R28R24 → P28R34.

*As for SURY, why the author choses these parameters namely bulk albedo, bulk emissivity, etc. as the output of SURY, this need to be clarified.*

Such a choice was made for making the SURY methodology generally applicable in existing bulk urban land-surface schemes. This information will be stated more clearly in the revised introduction, as shown in **bold** above.

*Surface-Area Index (SAI) is a crucial important factor in this paper to reparametrize the ground heat transport parameters, but why SAI is chosen to do this? Why these parameters need to be reparametrized?*

It is true that the SAI is an important parameter in the presented methodology. The physical reasoning for taking into acount this parameter (hence its importance) is given in section 2.1.1 (P5R1 → P5R29). This explanation will be better framed in the revised manuscript at the beginning of Section 2.1 as follows (starting at P4R1): *"In this section, the Semi-empirical URban canopY parametrization SURY is described. The translation of urban canopy parameters into urban bulk parameters takes into account the urban physical processes with regard to the ground-heat transport , the surface-radiation exchanges , and the surface-layer turbulent transport for momentum, heat and moisture :* ***The bulk thermal parameter values take into account the enhanced ground heat transport due to the increased contact surface with the***

*atmosphere (see Fortuniak, 2004) expressed by the Surface-Area-Index (SAI) in Section 2.1.1. Furthermore, the radiative bulk parameter values take into account the albedo reduction factor resulting from the radiative trapping by the urban canopy in Section 2.1.2. Finally, the enhanced surface drag on the wind by the buildings in the urban canopy take into account the building height in section 2.1.3.* *As a result, SURY introduces an efficient dependency of bulk urban land-surface schemes to the canopy parameters.* **Throughout the subsections below,** *the robustness of SURY is verified by comparing bulk parameters from top-down estimates with those translated from bottom-up urban canopy parameter inventories. Default values of the urban canopy parameters and those of the translated bulk parameters are determined. An overview of the urban canopy parameters (SURY input) and the bulk parameters (SURY output) is given in Table 1.*"

*Specific comments:*

*1. Page 4 Table 1: I think these parameters should be reworked because they are varied with different areas.*

The authors agree that the urban-canopy parmeters depend on the area under scope. As denoted in the introduction, these parameters are not always available in a consistent dataset, hence it is chosen to obtain and list a set of default parameter values derived from available datasets. In either way, we would be happy to learn how we could improve the parameter overview.

*2. Page 6 Equation 3: Please explain why use this equation to reparametrize the parameters.*

The formula can be obtained from geometrical considerations of an idealized parallel urban canyon with straight roads and flat roofs. The first term (1+ 2H/W) (1-R) represents the surface area index of the street canyon. In turn, it is subdivided in 1 x (1-R) which is the surface area of the street, and 2 H/W x (1-R) which is the surface area of the two walls in the street canyon. Finally, the second term R represents the surface area index of the roof.
This information will be added to the revised manuscript.

*3. Page 7 Equation 10 and 11:*
*These equations also need to be explained.*

In Equation 10, $\psi_{bulk}$ is the total albedo reduction factor of the urban canopy. The reduction factor is weighted according to the roof fraction R and the complementary street-canyon fraction (1-R). As stated before, flat roofs are considered, hence the roof fraction R does not lead to a albedo reduction. In contrast, multiple reflections take place for the street-canyon fraction (1-R) for which the canyon albedo reduction factor $\psi_{canyon}$ is taken into account expressed by Equation 11. As already stated in the manuscript (P7R19 and

further), equation 11 approximates the numerical estimation of Fortuniak (2007). This information will be supplemented to the revised manuscript.

*4. Page 8 Line 22: In my opinion the z0 is the most important parameter in surface layer turbulent fluxes parametrization, so I think at least z0 should be added in the sensitivity analysis.*

Agreed. As z0 (output of SURY) depends on the building height H (input of SURY) through Eq. 15, the sensitivity of the former is already covered by the 'EL' and 'EH' experiments.

*5. Page 29 Line 5: I think*
*the author should provide a website of the models.*

Thank you for this suggestion. We have made a public repository for SURY on Github under https://github.com/hendrikwout/sury. This information will be added to the manuscript. We will also provide a link to the project page of the modified version of the COSMO-CLM model with TERRA_URB that implements SURY.

---

## Referee Comment (RC2) · Anonymous Referee #2 · 25 May 2016

The paper by Wouters et al. presents the semi-emipirical urban canopy parametrization SURY and the urban bulk scheme TERRA-URB 2. SURY is used to derive bulk parameters from urban canopy parameters, which are used in more physically-explicit urban parametrization schemes. In this paper, TERRA-URB 2 with SURY parameters and coupled with the regional climate and weather model COSMO-CLM is evaluated with station and remote sensing data. Furthermore, a sensitivity analysis to SURY input parameters is conducted.

While the usage of SURY-derived parameters in conjunction with an urban bulk scheme

does not account for every detail represented by more explicit schemes, SURY greatly extends the applicability and transparency of bulk schemes.

The paper is well written and concise. The topic is highly relevant, thus I recommend publication after the following minor issues are addressed.

Page 12 line 4: The authors state that the range of the substrate albedo is derived from the range of the bulk albedo. From the description of SURY, I would expect exactly the opposite way of derivation: bulk albedo derived from the substrate albedo. Please clarify.

Page 24 line 13: The authors state that a lower roughness length resulted in lower wind speeds. I would expect higher wind speeds. This would be also in agreement with the reduced accumulation of excess heat in the urban centres.

I find Figure 6 quite confusing. For example, bulk parameters a given twice and the usage of space is not optimal. Maybe the authors can find a better representation of their work-flow.

Page 27 line 15: I suppose it should be "To this end" instead of "Therefore".

Throughout this paper, some citations miss parentheses, for example P2L5 and L23, P10L9 and L17.

---

## Referee Comment (RC3) · Anonymous Referee #3 · 27 May 2016

General comments:

This study evaluated a scheme for deriving bulk urban parameters from urban canopy parameters and then applying them in a meteorology model with a bulk urban model. The purpose is to better account for urban effects without the additional complication and computational burden of a more detailed street canyon model. I think that this approach is a good compromise between bulk models that designate parameters according to land use category and urban canopy models that require detailed urban morphology and high resolution grids (vertical and horizontal). However, its value over

existing bulk approaches would be greatly enhanced if there were a simple way to acquire the urban canopy parameters used by SURY for any user specified domain. If we are practically limited to using default values for the canopy parameters, the result is just a modified bulk model with little added value since it would not distinguish between different cities. Thus, it would be helpful to those of us who are tempted to use this approach if some additional guidance could be provided on how to easily acquire and process the needed urban canopy parameters.

While the paper clearly presents the study, I feel that the study has two main deficiencies. One is that the primary function of the SURY, which is to incorporate geographically specific urban canopy parameters into a simple bulk urban scheme is never really demonstrated. If it is too difficult for the developers of SURY to apply it to its full extent with actual canopy parameter data at the model grid resolution for a few cities in their domain, then it is unlikely that others would find it very useful. The other main deficiency is that the base model used in this study has significant errors in temperature simulation which obscures the evaluation of the urban parameterization and the sensitivity of the parameter uncertainty. I suggest that these deficiencies be addressed before publication.

Specific comments:

Page 6, ln4: what is SAI for natural land cover? Is it LAI?

Page 6, ln9-10: What is "this parameter"? I'm guessing that you are saying that the depth where the urban substrate changes to soil is equal to the building height h. Is this correct? If so, why should the substrate depth be equal to the building height? Please explain.

Page 18, ln9-10: These large biases in day and night LST and the under predicted diurnal range make it difficult to evaluate the urban model. How do you account for these errors in the base model when evaluating the UHI results?

Page 21, ln18-21: This discussion does not agree with Figure 5. It looks to me that the REF model underestimates the stable lapse rate between the lowest 2 observations at the rural site meaning it's less stable not "more stable". Figure 5 also shows that the UHI is underestimated near the ground due to the overestimation of the rural T.

Page 24, ln13: Why would lower roughness result in lower windspeed?

Page 28, ln13-23: This is a very important paragraph. As this paragraph points out, errors in the base model are obscuring the evaluation of the SURY and Urban scheme and the sensitivity analysis of the parameter uncertainty. Since, these errors undercut the value of this study it seems like some effort should have been made to reduce these errors.

Page 30, ln19: How is transpiration modeled? There should at least be a reference

Page 30, ln26: where does Fm come from?

Page 31, ln1: shouldn't rsa differ for heat and moisture?

Technical comments:

Table 1 caption last sentence typo: Hereby

Page 5, ln13: what is meant by lateral heat transport? "...within through..." doesn't make sense

Page 9, ln28: typo – Parater should be Parameter?

Page 10, ln3-4: This sentence in incomplete. It's missing a verb.

Page 13, ln32: typos – missing period after thermocouples, temperatuere is misspelled.

Page 21, ln3: Are the values given here for SUHI bias? Should say so.

Table 6: Are the values averaged vertically? Please explain what these mean. Also, I don't see an "R" column.

Page 28, ln18: "overwhelm" might be better than "overrule"

Page 28, ln24: "natre" should be "nature"

Page 31, ln7: This seems to be an errant line

Page 32, ln16: "withouth"

Page 33, ln7: should be 49.16 W m-2

---

## Author Comment (AC2) · 22 Jun 2016

[revised manuscript text omitted]

Optionally, a distinction is made between the albedo of roofs, roads and walls as follows:

$$\alpha_{\mathrm{bulk}} \simeq \frac{\left[ \alpha_{\mathrm{street,snow}} + 2\frac{H}{W}\alpha_{\mathrm{wall,snow}} \right]}{(1 + 2\frac{H}{W})} \psi_{\mathrm{canyon}}\left( \frac{H}{W} \right)(1-R) + \alpha_{\mathrm{roof,snow}} R \tag{14}$$

with

$$\alpha_{i,\mathrm{snow}} = (1 - f_{\mathrm{snow}})\alpha_i + f_{\mathrm{snow}}\alpha_{\mathrm{snow}}, \text{ for } i \text{ in (roof, wall, street)} \tag{15}$$

and where $\frac{\left[ \alpha_{\mathrm{street,snow}} + 2\frac{H}{W}\alpha_{\mathrm{wall,snow}} \right]}{(1 + 2\frac{H}{W})}$ is the averaged  albedo of the  streets and walls in the urban canyon. The effective infra-red emissivity $\epsilon_{\mathrm{bulk}}$ takes into account the same bulk albedo reduction factor $\psi_{\mathrm{bulk}}$ as follows:

$$\epsilon_{\mathrm{bulk}} = 1 - \psi_{\mathrm{bulk}}\left( 1 - \left( (1 - f_{\mathrm{snow}})\epsilon + f_{\mathrm{snow}}\epsilon_{\mathrm{snow}} \right) \
[revised manuscript text omitted]

---

## Author Comment (AC4) · 22 Jun 2016

The authors would like to thank the reviewer for reviewing the manuscript, and for their positive response by highlighting the added value of the manuscript. We are also thankful for their remarks for improving the manuscript. The responses to the comments can be found below, in which we refer to the revised manuscript containing the track changes, see http://www.geosci-model-dev-discuss.net/gmd-2016-58/gmd-2016-58-AC2-supplement.pdf.

Please note that some line-breaks are missing in the version with the track changes, a drawback of using latexdiff (mostly in combination with citations). Therefore, we also provide the new revised version without track changes www.geosci-model-dev-discuss.net/gmd-2016-58/gmd-2016-58-AC3-supplement.pdf.

A word of thanks will be provided in the next manuscript versions.

The paper by Wouters et al. presents the semi-emipirical urban canopy parametrization SURY and the urban bulk scheme TERRA-URB 2. SURY is used to derive bulk parameters from urban canopy parameters, which are used in more physically-explicit urban parametrization schemes. In this paper, TERRA-URB 2 with SURY parameters and coupled with the regional climate and weather model COSMO-CLM is evaluated with station and remote sensing data. Furthermore, a sensitivity analysis to SURY input parameters is conducted.

While the usage of SURY-derived parameters in conjunction with an urban bulk scheme does not account for every detail represented by more explicit schemes, SURY greatly extends the applicability and transparency of bulk schemes. The paper is well written and concise. The topic is highly relevant, thus I recommend publication after the following minor issues are addressed.

Page 12 line 4: The authors state that the range of the substrate albedo is derived from the range of the bulk albedo. From the description of SURY, I would expect exactly the opposite way of derivation: bulk albedo derived from the substrate albedo. Please clarify.

It is indeed so that SURY normally translates urban-canopy parameters (input) to bulk parameters (output). However, the parameter ranges from Stewart and Oke (2012) are those for the bulk parameters (alpha_bulk, lambda_bulk, Cv_bulk), and not for the substrate parameters (alpha, lambda_s, C_{v,s}). For clarity in future applications of SURY, we prefer to use only ranges for the input of SURY (ie., the urban-canopy parameters), which include the substrate parameters, not bulk parameters (output of SURY). Hence, for the sensitivity study, we reversed the equation of SURY to derive the substrate parameter ranges from the bulk parameters ranges in Stewart and Oke (2012), while keeping the other (morphological) parameters at their default values. In order to make this more clear for the reader, we make the following change to the revised manuscripts at P13R26-R31.

Page 24 line 13: The authors state that a lower roughness length resulted in lower wind speeds. I would expect higher wind speeds. This would be also in agreement with the reduced accumulation of excess heat in the urban centres.

Indeed, we have now replaced 'lower wind speed' with 'higher wind speeds'. In that case the reduced accumulation of excess urban heat and the lower temperature mentioned in the next sentence makes sense, indeed. We have changed this in the revised manuscript, see P27R13.

I find Figure 6 quite confusing. For example, bulk parameters a given twice and the usage of space is not optimal. Maybe the authors can find a better representation of their work-flow.

We have simplified the figure for making it more clear, see P29.

Page 27 line 15: I suppose it should be "To this end" instead of "Therefore".

We have replaced the text, see  P30R20

Throughout this paper, some citations miss parentheses, for example P2L5 and L23, P10L9 and L17.

We have checked and corrected the parentheses of the references throughout the manuscript.

---

## Author Comment (AC5) · 22 Jun 2016

The authors would like to thank the referee for the review of the manuscript and we appreciate their remarks and suggestions for improving the quality of the manuscript. The responses to the comments can be found below, in which we refer to the revised manuscript containing the track changes, see http://www.geosci-model-dev-discuss.net/gmd-2016-58/gmd-2016-58-AC2-supplement.pdf.

Please note that some line-breaks are missing in the version with the track changes, a drawback of using latexdiff (mostly in combination with citations). Therefore, we also provide the new revised version without track changes www.geosci-model-dev-discuss.net/gmd-2016-58/gmd-2016-58-AC3-supplement.pdf.

A word of thanks will be provided in the next manuscript versions.

General comments

This study evaluated a scheme for deriving bulk urban parameters from urban canopy parameters and then applying them in a meteorology model with a bulk urban model. The purpose is to better account for urban effects without the additional complication and computational burden of a more detailed street canyon model. I think that this approach is a good compromise between bulk models that designate parameters according to land use category and urban canopy models that require detailed urban morphology and high resolution grids (vertical and horizontal). However, its value over existing bulk approaches would be greatly enhanced if there were a simple way to acquire the urban canopy parameters used by SURY for any user specified domain. If we are practically limited to using default values for the canopy parameters, the result is just a modified bulk model with little added value since it would not distinguish between different cities. Thus, it would be helpful to those of us who are tempted to use this approach if some additional guidance could be provided on how to easily acquire and process the needed urban canopy parameters.

While the paper clearly presents the study, I feel that the study has two main deficiencies. One is that the primary function of the SURY, which is to incorporate geographically specific urban canopy parameters into a simple bulk urban scheme is never really demonstrated. If it is too difficult for the developers of SURY to apply it to its full extent with actual canopy parameter data at the model grid resolution for a few cities in their domain, then it is unlikely that others would find it very useful.

The authors agree that the final goal of the methodology is the application of detailed spatially-varying canopy-parameter datasets - distinguishing between the different residential, commercial and industrial areas - into existing bulk urban land-surface schemes.  Just like any other land-surface scheme including

the more complex explicit canyon models, the presented methodology is dependent on the availability of urban-canopy parameter (UCP) datasets. Many efforts for acquiring such parameter datasets already exist (as listed see below). The following types of datasets exist:

- Firstly, **detailed urban parameter inventories** exist for different campaigns over specific sites around the world (see e.g. the Preston site (Melbourne, Australia) in the Grimmond et al. 2011 Phase II Intercomparison paper). They are applicable for the specific urban terrain under scope (eg., applicable for offline urban climate modelling), but they do not include the city-wide variability

- Secondly, there are **detailed city-scale varying parameters**, but only for specific parameters and for specific cities, eg., CityGML 3D-urban canopy structure for Basel and Berlin (Schubert et al., 2013).

- Thirdly, **global datasets for urban-canopy parameters** exists, particularly that of Jackson et al. 2010 (based on site-specific parameter inventories worldwide). Based on 4 urban categories within 33 regions in the world, it provides information on the spatial extent, urban morphology, and thermal and radiative properties of building materials. Such datasets are intended for accounting for the urban-parameter variability on the global scales suitable for application in global climate modelling. Because their focus on the global scales, they do not to intend to deliver accuracy and detail on the scale of the cities needed for regional climate applications. In particular, the databases does not provide the variability in thermal and radiative parameters among the different urban classes and the additional spatial variability within one of the 33 region like Western Europe.

- Finally, the **local-climate zone classification (LCZ)** system (www.WUDAPT.org) aims to address these deficiencies. It provides recently developed tools (Stewart and Oke, 2012; Bechtel et al., 2015; See et al., 2015) for facilitating a coherent and detailed **urban canopy parameter dataset** with a world-wide coverage (more details can be found in the revised text at ...). However, such a dataset is currently under development. Specifically for the region under scope, the authors are currently involved in mapping the LCZs for the 3 largest Belgian cities (Ghent, Antwerp en Brussels) and are developing a new automated methodology to efficiently link these zones with morphological, radiative and thermal properties (Verdonck et al., submitted to Remote Sensing).

It is clear from the above that existing spatially-varying parameter datasets are currently under development, and this is particularly the case for the current evaluation region. Although the authors could not yet provide additional experiments that include more detailed spatially-varying UCP information, the authors are confident that our current study contains an important leap in urban climate research towards efficient and precise convection permitting **urban atmospheric modelling (Prein et al., 2015)**: The development of SURY anticipates on the ongoing UCP dataset advancements by making them applicable in existing bulk urban land-surface schemes. Hereby, the implementation procedure would be that UCPs are taken directly as input for

SURY translating them into bulk parameters for efficient urban atmospheric modelling. Therefore, the presented SURY framework will have wide applications in future studies noting the increasing interest and dataset development in the WUDAPT framework, the substantial amount of existing bulk schemes, the demand for efficiency and consistency in (ensemble) climate assessment and numerical weather prediction, and the need for more detailed parametrizations (Best and Grimmond, 2015). It should also be noted that the parameter sensitivity with SURY coupled to COSMO-CLM allowed us to make recommendations in the development of the UCP datasets.

As an intermediate solution, the current manuscript has developed a default set of UCPs in section 2.1, which combines SURY's theoretical framework, detailed existing urban-canopy parameter inventories, and modelling and observational studies.

We agree with the reviewer that the clarity 1) regarding the added value and future applications of SURY, and 2) about our recommendations with respect to the development of UCP datasets should be improved. Hereby, the SURY development should be better situated in existing literature (as explained above), and also the information about gathering and employing (upcoming) LCZ-based UCP datasets should be provided. Therefore, the authors propose to make the following text changes:

- in the introduction at P3R5-R31 (overview parameter sources), P4R3-R16 (added value SURY anticipating on more detailed parameter datasets)
- in the model setup: P13R15-R20 (motivating the use of the default parameter list).
- in the discussion and conclusions: P29R5-R19 (UCP application of SURY), P31R15-P33R3 (recommendations regarding the development of UCP datasets and their applications in atmospheric modelling) . Hereby, P32R1-R11 provides information about how to obtain more detailed datasets in future applications.
- and in the abstract: P1R17-P2R3

The other main deficiency is that the base model used in this study has significant errors in temperature simulation which obscures the evaluation of the urban parameterization and the sensitivity of the parameter uncertainty. I suggest that these deficiencies be addressed before publication.

For addressing this general comment above, we also take into account the following specific comment:

Page 28, ln13-23: This is a very important paragraph. As this paragraph points out, errors in the base model are obscuring the evaluation of the SURY and Urban scheme and the sensitivity analysis of the parameter uncertainty. Since, these errors undercut  the value of this study it seems like some effort should have been made to reduce these errors.

The authors are aware of the fact that general model performance may obscure the evaluation of SURY and parameter uncertainty. Hereby, the authors need to

stress that the current manuscript provides - to our best knowledge - one of the most comprehensive online evaluations with regard to the modelled urban heat islands, that both include BLUHI, CLUHI and SUHI. On the one hand, such a comprehensive analysis depicts the strengths of the coupled model system, eg., the ability to capture the diurnal and daily variability of the different urban heat islands. On the other hand, this extensive evaluation also reveals deficiencies, particularly, a general underestimation of the diurnal amplitude of nocturnal temperatures, urban heat islands and vertical temperature profile stability. In this respect, the authors agree with the reviewer that the SURY methodology is subject to deficiencies of the host model. However, such an undercutting could also happen for any other urban land-surface modelling strategy different from the SURY-methodology coupled to this or another host atmospheric model. In this respect, the URBMIP coupled model experiment (Trusilova et al., 2015) has shown that different urban land-surface parametrizations coupled to the same model share these similar issues.

Furthermore, addressing the underlying errors of the host atmospheric model is an enormous challenge. Hereby, fully-coupled atmospheric model systems deal with feedbacks between soil(-moisture) processes, atmospheric circulation, radiation, turbulence, cloud microphysics and land-atmosphere interactions. In the particular case of COSMO(-CLM), there are mainly two research communities (COSMO consortium and the CLM community) with over 200 people that are dealing with the improvement of the different atmospheric model components. Amongst others, recent efforts include the implementation of vegetation shading, improvements in the surface-layer and boundary-layer turbulence scheme, a new resistance formulation for bare soil evaporation and the improvement of cloud radiation coupling as a consequence of a recent published work by Brisson et al. (2016). For the current manuscript, the authors have tested different host model parameter set-ups, which could already improve the nocturnal boundary-layer stability and consequently the urban heat islands. This is particularly achieved by altering the settings of COSMOs boundary-layer turbulence scheme and taking into account soil-moisture conductivity (Schulz et al., 2016).

It should be noted that the urban parametrization in the COSMO-CLM model already provides an overall improvement regarding temperatures and urban heat islands (see P30R27-R28), particularly an alleviation of negative temperature biases in the urban areas (see P30R25-R28). As soon as other issues in the host atmospheric model are solved, additional benefits of SURY (and upcoming LCZ-based UCP datasets) will come forward automatically. Our previous study Wouters et al. (2013) indeed shows that general model improvement interacts with urban-climate modelling performance: Herein, it was shown with an idealized boundary-layer model that the model representation of the boundary-layer (directly affected by surface-atmospheric interaction) is important for well-capturing the urban heat island. For instance, an overall underestimation of nocturnal stability found in the host model gives rise to an underestimated nocturnal canopy-layer urban heat island. We agree with the reviewer that the context regarding the urban-climate modelling performance should be addressed in more detail in the text. Therefore, we now

provide a separate paragraph at P30R27-P31R4.

Despite the model errors, the authors are confident that the development of SURY and the online evaluation and sensitivity study provides a substantial added value to existing literature: it has enabled us to formulate recommendations for more precise urban-climate modelling at the convection permitting scales (hence the SURY methodology coupled to COSMO-CLM model features an efficient test bed):  Particularly, the sensitivity of the UCPs relative to the overall model errors indicates the (relative) importances of both the ongoing advancements in the urban canopy parameters (see above) and in the atmospheric model system for more reliable urban climate modelling. In order to make our recommendations more clear for the reader, the following changes are made to the manuscript:

- in the results section: P17R24-R25 and P24R17-R19 (model performance)
- in the discussion and conclusions: P30R27-P31R4 (discussion about atmospheric model performance), P32R21-P33R3 (recommendations regarding atmospheric model improvement and urban-canopy parameter improvements)
- and in the abstract: P1R13-P2R3

Specific comments:
Page 6, ln4: what is SAI for natural land cover? Is it LAI?

SAI refers to the The Surface Area Index, which is defined as the ratio between the land-surface area - that envelops the urban canopy -  and the plan area. The definition can be found at P5R29-P6R2

Page 6, ln9-10: What is "this parameter"? I'm guessing that you are saying that the depth where the urban substrate changes to soil is equal to the building height h. Is this correct? If so, why should the substrate depth be equal to the building height? Please explain.

Indeed, 'this parameter' refers to the thickness of the 'urban substrate layer'. The latter is introduced for representing thermal properties of the urban canopy in thermal contact with the natural soil layer below. This represents  the (thermal mass of the) buildings and as such its thickness equals the building height H. In order to make the formulation and definitions more clear, we have revised the corresponding paragraph (see P7R24-P8R18)

Page 18, ln9-10: These large biases in day and night LST and the under predicted diurnal range make it difficult to evaluate the urban model. How do you account for these errors in the base model when evaluating the UHI results?

As clear from Fig. 4, biases are of similar magnitude for the different urban classes and the rural class. As a result, the SUHI, calculated as the difference

between each urban class and the rural class, is well reproduced by the model compared to the observed SUHI, as indicated in the text.

Page 21, ln18-21: This discussion does not agree with Figure 5. It looks to me that the REF model underestimates the stable lapse rate between the lowest 2 observations at the rural site meaning it's less stable not "more stable". Figure 5 also shows that the UHI is underestimated near the ground due to the overestimation of the rural T.

It is true that the model shows less stable lapse rates than the observed lapse rate. However,  the model is still able to reproduce the contrast between the urban and rural site, ie. the more stable boundary layer in the rural site compared to the industrial site. We now state this more clearly in the text: P24R17-R19

Hereby, it should be noted that the mentioned model issues with the boundary-layer stability are indicated at P24R14-P25R17 and P30R31-R32

Page 24, ln13: Why would lower roughness result in lower windspeed?

Indeed, 'lower wind speed' should be replaced with 'higher wind speed', which is now changed in the revised manuscript (see P27R13). In that case,  the reduced accumulation of excess urban heat and the lower temperature mentioned in the next sentence also makes sense.

Page 28, ln13-23: This is a very important paragraph. As this paragraph points out, errors in the base model are obscuring the evaluation of the SURY and Urban scheme and the sensitivity analysis of the parameter uncertainty. Since, these errors undercut the value of this study it seems like some effort should have been made to reduce these errors.

This comment is addressed together with the general comments. See above.

Page 30, ln19: How is transpiration modeled? There should at least be a reference

This is added at P35R5-R8

Page 30, ln26: where does Fm come from?

Fm (maximal  moisture flux that the soil can sustain) is adopted from formulation of Dickinson (1984). The reference is added to the revised manuscript at P35R15

Page 31, ln1: shouldn't rsa differ for heat and moisture?

The COSMO-CLM model considers the transfer resistance for 'scalars', which

refers to both 'heat' and 'moisture'. This was made more clear at line P35R17

Technical comments:

We will adopt the technical comments in the revised manuscript. Where necessary, explanations are given below.

Table 1 caption last sentence typo: Hereby

Agreed. See P6

Page 5, ln13: what is meant by lateral heat transport? ". . .within through. . ." doesn't make sense

The sentence is reformulated at: P6R4-R5.

Page 9, ln28: typo – Parater should be Parameter?

See P12R3

Page 10, ln3-4: This sentence in incomplete. It's missing a verb.

'according to' is replaced by 'is according to', see P12R14

Page 13, ln32: typos – missing period after thermocouples, temperatuere is mis spelled.

See P16R25

Page 21, ln3: Are the values given here for SUHI bias? Should say so.

Indeed. This is now indicated at P24R2.

Table 6: Are the values averaged vertically? Please explain what these mean. Also, I don't see an "R" column.

Yes, the values are vertically averaged. The revised text can be found at the figure caption on P25.

Page 28, ln18: "overwhelm" might be better than "overrule"

We agree that "overrule" is misused. We replaced it with "exceed", see P32R21-R27

Page 28, ln24: "natre" should be "nature"

see P33R4

The line has been removed from the manuscript, see P35R23

see P37R2

see P37R28

---

## Author Comment (AC6) · 22 Jun 2016

In addition to our previous replies in black font below, we have now included an additional red font that provides additional information and points to the location of the changes in the revised manuscript. The latter can be found at http://www.geosci-model-dev-discuss.net/gmd-2016-58/gmd-2016-58-AC2-supplement.pdf.

Please note that some line-breaks are missing in the version with the track changes, a drawback of using latexdiff (mostly in combination with citations). Therefore, we also provide the new revised version without track changes www.geosci-model-dev-discuss.net/gmd-2016-58/gmd-2016-58-AC3-supplement.pdf.

The authors would like to thank the referee for the review of the manuscript. We appreciate the suggestions for clarifying the manuscript in some of its key points.  As part of the interactive discussion of GMDD, we provide a reply to the different reviewer's comments. A definitive answer will be provided at the time of the revision, in which the necessary changes will be indicated in the revised manuscript.

*General comments:*

*This paper boasts to present a Semiempirical URban-canopY parametrization SURY, which bridges the gap between bulk urban land-surface schemes and explicit-canyon schemes. But it lacks the comparison between the bulk urban schemes and explicit-canyon schemes....*

The authors agree that model intercomparison studies comparing bulk schemes on the one hand and the explicit-canyon schemes on the other hand are important to substantiate the development and advantages of SURY as an 'in-between' model approach.  In this respect, *the introduction in our paper summarizes a qualitative comparison between the bulk schemes and the explicit canyon schemes (see P2R9→P3R8). For the revised manuscript, we will add the references to studies that compare the different schemes as follows (at P2R10): "Even though their purpose of representing urban physics in land-surface schemes of atmospheric models is the same,* **intercomparison studies (Grimmond et al., 2011; Best and Grimmond, 2015; Trusilova et al.2015; Karsisto et al., 2015) demonstrate** *that they differ in terms of modelling strategy, complexity, input parameters and applicability"*

see P2R15  [or P2R9 in the version without the track changes]

Furthermore, the intercomparison studies support our model development for bringing canopy-dependent urban physics to existing urban bulk land-surface schemes (see in bold just below), also allowing to consolidate urban canopy parameter datasets with urban bulk parameters (as stated in our conclusions). Therefore, the following information is to the revised introduction as well (at

P2R29): *"At first, the urban canopy parameters, which include information about* **the three-dimensional** *urban morphology and material properties, are obtained from detailed inventories (Loridan and Grimmond, 2012; Jackson et al., 2010).* **The first urban model intercomparison project demonstrate that such parameter information is important for improved modelling performance in existing urban land-surface schemes (Best and Grimmond, 2015)."**

*This is covered in the revised introduction paragraph: P3R9-R27*

*Finally, a full quantitative comparison between urban land-surface schemes, which is covered by existing studies as mentioned above, is outside of the scope of this paper about the specific development of SURY.*

*... So, I question whether the SURY scheme is necessary or not.*

The authors agree with the reviewer that the added value of SURY over existing methodologies needs to be stated very clear in the manuscript. Therefore, we will include the following information as bullet-points in the revised introduction (at P3R10):
- (*as already stated in the introduction*) Based on detailed observational studies, modelling experiments and available parameter inventories, SURY represents a robust translation of urban-canopy parameters containing three-dimensional information towards bulk parameters.
- the translation allows to combine advantages (hence bridges the gap) of both bulk schemes and explicit-canyon schemes in urban modelling studies. Especially, **it brings canopy-dependent urban physics** (used to be reserved for the explicit-canyon schemes before) **to the existing urban bulk land-surface schemes.** This could be done while preserving the low computational cost and low complexity of the bulk schemes.
- the translation offers versatility and consistency in choosing between urban-canopy parameters from bottom-up inventories and bulk parameters from top-down inventories.

This is covered more explpicitely at P4R3-R17

Note that a more extensive discussion about advantages (and limitations) of SURY with regard to applicability, versatility, model consistency and the computational cost  is provided in the 'discussion and conclusions'-section, see R26R1→R27R14 and R28R24 → P28R34.

This is now covered more clearly in the revised manuscript at P29R5-P30R19 and P33R4-R18

*As for SURY, why the author choses these parameters namely bulk albedo, bulk emissivity, etc. as the output of SURY, this need to be clarified.*

Such a choice was made for making the SURY methodology generally applicable in existing bulk urban land-surface schemes. This information will be stated more clearly in the revised introduction, as shown in **bold** above.

*Surface-Area Index (SAI) is a crucial important factor in this paper to reparametrize the ground heat transport parameters, but why SAI is chosen to do this? Why these parameters need to be reparametrized?*

It is true that the SAI is an important parameter in the presented methodology. The physical reasoning for taking into acount this parameter (hence its importance) is given in section 2.1.1 (P5R1 → P5R29). This explanation will be better framed in the revised manuscript at the beginning of Section 2.1 as follows (starting at P4R1): *"In this section, the Semi-empirical URban canopY parametrization SURY is described. The translation of urban canopy parameters into urban bulk parameters takes into account the urban physical processes with regard to the ground-heat transport , the surface-radiation exchanges , and the surface-layer turbulent transport for momentum, heat and moisture : **The bulk thermal parameter values take into account the enhanced ground heat transport  due to the increased contact surface with the atmosphere (see Fortuniak, 2004) expressed by the Surface-Area-Index (SAI) in Section 2.1.1. Furthermore, the radiative bulk parameter values take into account the albedo reduction factor resulting from the radiative trapping by the urban canopy in Section 2.1.2. Finally,  the enhanced surface drag on the wind by the buildings in the urban canopy take into account the building height in section 2.1.3.** As a result, SURY introduces an efficient dependency of bulk urban land-surface schemes to the canopy parameters. **Throughout the subsections below,** the robustness of SURY is verified by comparing bulk parameters from top-down estimates with those translated from bottom-up urban canopy parameter inventories. Default values of the urban canopy parameters and those of the translated bulk parameters are determined. An overview of the urban canopy parameters (SURY input) and the bulk parameters (SURY output) is given in Table 1."*

*The revised text can be found in the revised manuscript at P5R8-R20*

*Specific comments:*

*1. Page 4 Table 1: I think these parameters should be reworked because they are varied with different areas.*

The authors agree that the urban-canopy parameters depend on the area under scope. As denoted in the introduction, these parameters are not always

available in a consistent dataset, hence it is chosen to obtain and list a set of default parameter values derived from available datasets.

More particularly, the authors agree that the methodology should employ more detailed spatially-varying canopy-parameter datasets - distinguishing between the different residential, commercial and industrial areas - into existing bulk urban land-surface schemes. Just like any other land-surface scheme including the more complex explicit canyon models, the presented methodology is dependent on the availability of urban-canopy parameter (UCP) datasets. Many efforts for acquiring such parameter datasets already exist (as listed see below). The following types of datasets exist:
- Firstly, **detailed urban parameter inventories** exist for different campaigns over specific sites around the world (see e.g. the Preston site (Melbourne, Australia) in the Grimmond et al. 2011 Phase II Intercomparison paper). They are applicable for the specific urban terrain under scope (eg., applicable for offline urban climate modelling), but they do not include the city-wide variability
- Secondly, there are **detailed city-scale varying parameters**, but only for specific parameters and for specific cities, eg., CityGML 3D-urban canopy structure for Basel and Berlin (Schubert et al., 2013).
- Thirdly, **global datasets for urban-canopy parameters** exists, particularly that of Jackson et al. 2010 (based on site-specific parameter inventories worldwide). Based on 4 urban categories within 33 regions in the world, it provides information on the spatial extent, urban morphology, and thermal and radiative properties of building materials. Such datasets are intended for accounting for the urban-parameter variability on the global scales suitable for application in global climate modelling. Because their focus on the global scales, they do not to intend to deliver accuracy and detail on the scale of the cities needed for regional climate applications. In particular, the databases does not provide the variability in thermal and radiative parameters among the different urban classes and the additional spatial variability within one of the 33 region like Western Europe.
- Finally, the **local-climate zone classification (LCZ)** system (www.WUDAPT.org) aims to address these deficiencies. It provides recently developed tools (Stewart and Oke, 2012; Bechtel et al., 2015; See et al., 2015) for facilitating a coherent and detailed **urban canopy parameter dataset** with a world-wide coverage (more details can be found in the revised text at ...). However, such a dataset is currently under development. Specifically for the region under scope, the authors are currently involved in mapping the LCZs for the 3 largest Belgian cities (Ghent, Antwerp en Brussels) and are developing a new automated methodology to efficiently link these zones with morphological, radiative and thermal properties (Verdonck et al., submitted to Remote Sensing).

It is clear from the above that existing spatially-varying parameter datasets are currently under development, and this is particularly the case for the current evaluation region. The development of SURY anticipates on the ongoing UCP dataset advancements by making them applicable in existing bulk urban land-surface schemes. As an intermediate solution, the current manuscript has developed a default set of UCPs in section 2.1 (table 1), which combines SURY's theoretical framework, detailed existing urban-canopy parameter inventories, and modelling and observational studies. More detailed spatially-varying urban-canopy parameters can be employed as soon as they become available.

In order to integrate this information more clearly in the manuscript, the authors propose to make the following text changes:

- in the introduction at P3R5-R31 (overview parameter sources), P4R3-R16 (added value SURY anticipating on more detailed parameter datasets)
- in the model setup: P13R15-R20 (motivating the use of the default parameter list).
- in the discussion and conclusions: P29R5-R19 (UCP application of SURY), P31R15-P33R3 (recommendations regarding the development of UCP datasets and their applications in atmospheric modelling)
- and in the abstract: P1R17-P2R3

*2. Page 6 Equation 3: Please explain why use this equation to reparametrize the parameters.*

The formula can be obtained from geometrical considerations of an idealized parallel urban canyon with straight roads and flat roofs. The first term (1+ 2H/W) (1-R) represents the surface area index of the street canyon. In turn, it is subdivided in 1 x (1-R) which is the surface area of the street, and 2 H/W x (1-R) which is the surface area of the two walls in the street canyon. Finally, the second term R represents the surface area index of the roof.

It is added at P7R10-R21

*3. Page 7 Equation 10 and 11:*
*These equations also need to be explained.*

In Equation 10, $\psi\_bulk$ is the total albedo reduction factor of the urban canopy. The reduction factor is weighted according to the roof fraction R and the complementary street-canyon fraction (1-R). As stated before, flat roofs are considered, hence the roof fraction R does not lead to a albedo reduction. In contrast, multiple reflections take place for the street-canyon fraction (1-R) for which the canyon albedo reduction factor $\psi\_canyon$ is taken into account expressed by Equation 11. As already stated in the manuscript (P7R19 and

further), equation 11 approximates the numerical estimation of Fortuniak (2007). This information will be supplemented to the revised manuscript.

*See P9R20-R24*

*4. Page 8 Line 22: In my opinion the z0 is the most important parameter in surface layer turbulent fluxes parametrization, so I think at least z0 should be added in the sensitivity analysis.*

Agreed. As z0 (output of SURY) depends on the building height H (input of SURY) through Eq. 15, the sensitivity of the former is already covered by the 'EL' and 'EH' experiments.

*See P27R12-R17*

*5. Page 29 Line 5: I think*
*the author should provide a website of the models.*

Thank you for this suggestion. We have made a public repository for SURY on Github under https://github.com/hendrikwout/sury and added this information to the manuscript. Furthermore, this new section now also provides a link to the project page of the modified version of the COSMO-CLM model with TERRA_URB that implements SURY.

The additional information can be found at P33R20-R24

---

## Author Comment (AC7) · 22 Jun 2016

Dear chief editor, Dear Prof. Astrid Kerkweg,

The authors would like to thank you for this comment. According to your suggestion, we have changed the title for indicating the development SURY:

"The efficient urban-canopy dependency parametrization SURY (v1.0) for atmospheric modelling: description and application with the COSMO-CLM model (v5.0_clm6) for a Belgian summer"

Yours sincerely, Dr. Hendrik Wouters (on behalf of the co-authors)

---

## Referee Report (RR1)

**General comments:**
This study proposed the use of a Semi-empirical URban-canopY
parametrization (SURY) to improve urban simulations. The idea is to better
represent urban effects parameterized in bulk urban land surface schemes
without employing computational expensive explicit-canyon schemes.
Simulation results were evaluated by both in-situ and remote sensing
measurements, and the proposed approach was able to improve urban heat
island simulated by a bulk urban model.

While the use of SURY shows improvements in urban simulations, there is
one thing that needs to be addressed in this manuscript.
The proposed SURY does not really carry out the heterogeneous features
presented in explicit-canyon schemes, and there is no comparison between
the simulation results from SURY and explicit-canyon schemes. Therefore,
it is hard to tell if SURY is able to reasonably serve as an alternative of
explicit-canyon schemes with lower computational cost. I suggest that this
deficiency be addressed before publication.

**Specific comments:**
Page 19, ln 8-10: The authors claimed that the temperature overestimation
for the rural site is larger for REF than for STD, because of the advection of
excess heat from urban areas towards the rural areas. However, there was no
result supporting the occurrence of heat "advection" at that time. Why the
overestimation is due to heat advection but not the changes with SURY?

Page 24, ln 1-2: Similar issue as the previous one. Is there any result
suggesting the occurrence of heat advection at that time?

---

## Author Response (AR2)

Dear editor,
Dear Prof. Min-Hui Lo,

We would like to thank you for taking care of the review process and for your positive response. We send our next revision of our manuscript *"The efficient urban-canopy dependency parametrization SURY (v1.0) for atmospheric modelling: description and application with the COSMO-CLM model (v5.0_clm6) for a Belgian summer"*. We have taken into account the additional comments of the reviewers and are confident that the manuscript is ready for final publication in GMD.

Our replies to the author's comments can be found on the next pages below. As before, we provide a version with track changes with respect to the previous revision at the end of this document.

Finally, we would like to thank all the referees for their helpful suggestions.

We are hoping for your positive response.

Yours sincerely,
Dr. Hendrik Wouters

On behalf of the co-authors.

**Response to Referee #2 (Report #1)**

The authors would like to thank the referee for his/her positive response to the submission of the manuscript. The manuscript with track changes since the previous version can be found at the end of this document.

**Response to Referee #4 (Report #2)**

The authors would like to thank the referee for the review of the manuscript and we appreciate his/her remarks on the manscript and suggestions for improving the quality of the manuscript. Our answers can be found below, in which we refer to the revised manuscript with track changes at the bottom of this document

General comments

This study proposed the use of a Semi-empirical Urban-canopY parametrization (SURY) to improve urban simulations. The idea is to better represent urban effects parameterized in bulk urban land surface schemes without employing computational expensive explicit-canyon schemes. Simulation results were evaluated by both in-situ and remote sensing measurements, and the proposed approach was able to improve urban heat island simulated by a bulk urban model.

While the use of SURY shows improvements in urban simulations, there is one thing that needs to be addressed in this manuscript. The proposed SURY does not really carry out the heterogeneous features presented in explicit-canyon schemes, ...

It is true that SURY does not serve to represent the full heterogeneinity of the urban canopy as done in more complex schemes. Instead, SURY aims for translating urban canopy parameters into bulk parameters. In this way, SURY introduces the dependency of urban canopy parameters (used to be preserved to explicit-canyon schemes) to bulk parametrizations while preserving their low computational cost. A more extensive description of the novelty of SURY can be found in the abstract (see P1R1-R11) and the introduction (see P3R27-P4R5).

... and there is no comparison between the simulation results from SURY and explicit-canyon schemes.

It is true that no comparison to other parameterizations has been given. Instead, the focus of the paper is the SURY's methodology and the application with the COSMO-CLM model including the urban-canopy parameter sensitivity study. Systematic comparisons between bulk parametrizations employing (or without) SURY and other urban (explicit-canyon) parametrizations should be addressed in future intercomparison studies (such as Karsisto et al., 2015). In fact, an intercomparison study of TERRA_URB/SURY with explicit-canyon schemes for an urban site in Singapore is currently in preparation by Demuzere et al. (2016). The results indicate that all models included in this intercomparison (Community Land Model – Urban, SURFEX (TEB), SUEWS and

TERRA_URB/SURY) show comparable skill.

Therefore, it is hard to tell if SURY is able to reasonably serve as an alternative of explicit-canyon schemes with lower computational cost. I suggest that this deficiency be addressed before publication.

The authors agree that SURY does not provide an alternative for explicit canyon schemes when aiming for resolving the full heterogeneity in the urban canopy. Still, we are confident that SURY will have applications in future numerical weather prediction and regional climate studies, given the available urban bulk parametrizations and SURY's several advantages as described in the 'discussion and conclusions' section (see P28R12-P29R25). In order to situate this better in the manuscript, we have reformulated the last paragraph, see P31R29-P32R16.

Page 19, ln 8-10: The authors claimed that the temperature overestimation for the rural site is larger for REF than for STD, because of the advection of excess heat from urban areas towards the rural areas. However, there was no result supporting the occurrence of heat "advection" at that time. Why the overestimation is due to heat advection but not the changes with SURY?

Page 24, ln 1-2: Similar issue as the previous one. Is there any result suggesting the occurrence of heat advection at that time?

SURY only introduces modifications to the surface-atmospheric interaction formulation in urban areas. Therefore, any changes in the model results regarding the rural areas can only be the result of a propagation of urban effects through the atmosphere. In the case of heating in rural areas, it is postulated that such propagation is established by advection of excess urban heat from the urban areas towards the rural areas. Even though it is dealt with an indirect effect, it should be noted that it is still the result of the urban land-surface scheme. In order to make it more specific, we have reformulated the respective sentences at P20R6-R7 and P25R1-R2.

Demuzere, M., Harshan, S., Jarvi, L., Roth, M., Grimmond, C.S.B, Masson, V., K.W. Oleson, Velasco, E., Wouters, H, 2016. Impact of urban canopy models and external parameters on the modelled urban energy balance. In preparation for Quarterly Journal of the Royal Meteorological Society.

Karsisto, P., C. Fortelius, M. Demuzere, C. S. B. Grimmond, K. W. Oleson, R. Kouznetsov, V. Masson, and L. Järvi. 2015. "Seasonal Surface Urban Energy Balance and Wintertime Stability Simulated Using Three Land-Surface Models in the High-Latitude City Helsinki." Quarterly Journal of the Royal Meteorological Society, n/a – n/a. doi:10.1002/qj.2659.

**Response to Referee #5 (Report #3)**

The authors would like to thank Dr. Yi-Ying Chen for reviewing the manuscript and for his positive response. We also appreciate the remarks and suggestions for improving the quality of the manuscript. Our answers can be found below, in which we refer to the revised manuscript with track changes at the bottom of this document.

Overall, the authors present a detail model description of SURY and it would be useful for utilizing the urban canopy within climate models with a better representation of land-atmosphere energy exchanges, which improves the capability of climate models or regional weather models to capture the heat island observation. This manuscript also demonstrated that the model can reproduced both SUHI and CLUHI at two selected sites in Belgium, and the work can be extend to the global scale when the global urban dataset from WUDAPT.org is ready. I suppose that this manuscript can be published in GMD library. However, I found a few minor issues which might need to be addressed. Below please see my comments on this manuscript.

General comment:
My primary concerns with this manuscript are the way that SURY applied for resolving the surface temperature (top layer soil temperature) and the physical representation of the rainfall interception loss within the urban-canopy model.
For my first concern, SURY re-parameterized the surface characteristic of urban land-cover by making use of a bulk approximation within an existing canopy model, which is based on a signal layer "Big-leaf" assumption. Thus, the physical representation of the surface temperature of the Eq. (1) can be treated as a lump land surface temperature for several tiles including imperious surface, canopy cover, bare soil or grass land. Under this assumption, we might be able to apply the surface energy balance equation to resolve the surface temperature (top layer soil temperature) for different land covers in a modeled grid. Thus, different land-cover tiles share the same surface temperature, which causes the difficulty to validate the model calculated land surface temperature.

It is true that, in principle, one could treat the different land-cover types with a single ground temperature profile (in a single tile) with the presented methodology. However, in the current implementation with TERRA_URB in the COSMO-CLM model (see appendix A6), a separation is made between the urban canopy (impervious land-cover) and the natural land (bare soil, vegetated land-cover…). Hereby, the two ground tiles are resolved separately (see appendix A5). This way, the urban and natural land-cover tiles provide distinct land surface temperatures (which is also the case for ground temperature and moisture profiles below). The surface temperatures have been validated with LST from satellite data by making and weighting the results of the respective tiles.

Apart from the issue of validating the model predicted land surface temperature, the bulk surface heat conductivity ($\lambda_{bulk,s}$) was parameterized through a linear approximation by multiplying the original soil heat conductance ($\lambda_s$) with the SAI value (see Eq. (9)). But, in fact, the $\lambda_s$ itself is a function of the top layer soil moisture content. In other words, the physical characteristic for the building is somewhat likes a permeable surface layer. Is this assumption reasonable?

In principle, one could have (partially) permeable surface for the urban canopy with the presented methodology. This could be applicable for roads that are not entirely petrified, for example. This is, however, not considered in the current model implementation, meaning that urban canopy tile (resolved separately, see above) is entirely impervious. In this respect, $\lambda_s$ for the urban canopy is not a function of the soil moisture content. For details, see P36R10 and further.

Besides, I was confused about the way for solving land surface temperature (top layer soil surface temperature) in the TERRA_URB/SURY. Does the model incorporate the anthropogenic heat emission to the surface energy budget to resolve the surface temperature? Or it's just a simple source or sink term in the surface energy budget and the land surface received the feedback energy from incoming long-wave radiation.

TERRA_URB indeed considers the anthropogenic heat emission. Hereby, the anthropogenic heat emission is added to the turbulent heat release to the atmosphere, hence acts as a simple source term to the first atmospheric model layer. This way, the land surface (cfr. The surface energy budget, surface temperature...) is indirectly affected. In order to make this more clear, we have reformulated the last sentence of appendix A4, see P35R21-R22.

For the second comment, canopy can intercept 15% to 85% of total precipitation and can produce a substantial fast feedback to the atmospheric rainfall. SURY used a bulk average building height (H) to re-parameterize the original surface roughness ($z_0$ in Eq. (7)). I can't find the any equation/information regarding to the parameterization of the urban-canopy interception (Eimp) in the manuscript. The authors selected a drying period (sunny days) to test the newly proposed parameterization approach (SURY) may not have any impact on the result of the model simulation by making use of the current model configuration. However, the heat island effect can trigger or reinforce the afternoon thunderstorm events. Therefore, it is necessary to improve the physical representation of water budget in the urban-canopy model. I suggest that the authors improve the model description regarding to the urban-canopy water budget (see Eq. A12), the works on re-parametrization of the wet urban-canopy interception loss (Eimp in Eqs. (A6 and A12) will be appreciated. Besides, evaporative cooling effect from the urban-canopy loss can be studied or quantified. Anyway, the above issue seems outside the scope of this study, it will only become a substantial issue for large scale model simulations, such as long-term climate projections or weather predictions.

It is true that urban-canopy interception (Eimp) can be an important term

during rainfall periods. As stated in the end of appendix A2, the calculation of Eimp is covered in the next appendix (A3), particularly by equations A10 and A11. In order to make this more clear, we have reformulated the last line in appendix A2 (see P34R20).

Particularly, the impact of impervious water storage parametrization on urban climate modelling was the main topic of our previous study that has been published recently (Wouters et al., 2015). Hereby, the (offline) effect of water-storage parametrization on the urban energy balance and surface temperatures throughout has been quantified. In principle, it should be straightforward to address the evaporative cooling effect from the urban-canopy loss on the atmospheric fields (or more generally, the impact of urbanization on the water cycle) with the coupled atmospheric modelling setup employed in this study.

Specific comment:
P6L6 and P32L12:
Within the Eq. (1), the symbol T is the surface temperature. If this description is correct, the notation of "T" may have a conflict to the "T" in the Eq. (A7). Please make a change for the symbol of "T" in the Eq. (1) or Eq. (A7).

We have replaced T in Eq. (A6 (in the revised manuscript)) by T_r for indicating vegetation (see P34R3).

P7L3 and P32L31:
The definition of symbol "H" in the main context conflicts with the definition of symbol "H" in the appendix (see Eq. (A7)).

We have replaced all *H*-occurrences for building height with the small letter *h, including the equations throughout the text.*

P32L32:
Please remove the empty space line in front of Eq. (A8)

We have removed the empty equation (see P33R10-R11); please note that the equation in red is removed in the final manuscript. Hereby, the numbering of the subsequent equations has been shifted.

**Response to Referee #3 (Report #4)**

The authors would like to thank the referee for reviewing the revision of the manuscript and for his/her positive response. We would also like to thank, once again, for his/her previous suggestions which have substantially contributed to the overall quality of the manuscript. The revision of the manuscript with track changes since the previous version can be found at the bottom of this document.

Technical edits:

P2ln28: "Moreover, (Best and Grimmond, 2015) have shown…" should be: "Moreover, Best and Grimmond (2015) have shown…"

P29ln25: "between between"

Thank you for the additional comments. We have made the corrections to the manuscript.

[revised manuscript text omitted]

15  $$C_{v,\mathrm{bulk}}(z) = C_{v,\mathrm{soil}}, \text{ for } z \geq \underline{H}h \tag{6}$$

An analogous formulation is considered for the vertical profile of the bulk heat conductivity $\lambda_{\mathrm{bulk}}(z)$:

$$\lambda_{\mathrm{bulk}}(z) = \left(1 - \frac{z}{H}\frac{z}{h}\right) \lambda_{\mathrm{bulk,s}} + \frac{z}{H}\frac{z}{h} \lambda_{\mathrm{soil}}, \text{ for } z < \underline{H}h \tag{7}$$

$$\lambda_{\mathrm{bulk}}(z) = \lambda_{\mathrm{soil}}, \text{ for } z \geq \underline{H}h \tag{8}$$

where $\lambda_{\mathrm{soil}}$ is the heat conductivity of the natural soil, and $\lambda_{\mathrm{bulk,s}}$ is the bulk surface heat conductivity:

20  $$\lambda_{\mathrm{bulk,s}} = \lambda_s\,\mathrm{SAI}. \tag{9}$$

 The default urban canopy parameters  (see also Table 1) are set equal to the recommended values for the medium urban  category in Loridan and Grimmond (2012), see their Table IV (stage 5b): Herein, the height of the buildings  $h$, the canopy height-to-width ratio $\frac{h}{w_c}$  and the roof fraction $R$ are  equal to $15\,\mathrm{m}$, 1.5 and 0.667, respectively. According to Equation 3, the values for $R$ and $\frac{h}{w_c}$ lead to an SAI of 2.0. The

$$\mu_{\mathrm{bulk,s}} = \sqrt{\lambda_{\mathrm{bulk,s}} C_{v,\mathrm{bulk,s}}}.$$

 default values for the surface heat conductivity $C_{v,s}$ ($1.25 \times 10^6\,\mathrm{J\,m^{-3}\,K^{-1}}$) and the surface heat capacity $\lambda_s$ ($0.767\,\mathrm{W\,m^{-1}\,K^{-1}}$) are the respective weighted averages from the values for roof ($C_{v,\mathrm{roof}} = 1.2 \times 10^6\,\mathrm{J\,m^{-3}\,K^{-1}}$; $\lambda_{\mathrm{roof}} = 0.4\,\mathrm{W\,m^{-1}\,K^{-1}}$), wall ($C_{v,\mathrm{roof}} = 1.2 \times 10^6\,\mathrm{J\,m^{-3}\,K^{-1}}$; $\lambda_{\mathrm{wall}} = 1.0\,\mathrm{W\,m^{-1}\,K^{-1}}$) and road ($C_{v,\mathrm{road}} = 1.5 \times 10^6\,\mathrm{J\,m^{-3}\,K^{-1}}$; $\lambda_{\mathrm{road}} = 0.8\,\mathrm{W\,m^{-1}\,K^{-1}}$). Hereby, the weighted averages are calculated according to the surface fractions of roofs, walls and roads in the urban canopy:

$$C_{v,s} \approx \frac{1-R}{\mathrm{SAI}}\left(2\frac{h}{w_c}C_{v,\mathrm{wall}} + C_{v,\mathrm{road}}\right) + \frac{R}{\mathrm{SAI}}C_{v,\mathrm{roof}} \tag{10}$$

$$\lambda_s \approx \frac{1-R}{\mathrm{SAI}}\left(2\frac{h}{w_c}\lambda_{\mathrm{wall}} + \lambda_{\mathrm{road}}\right) + \frac{R}{\mathrm{SAI}}\lambda_{\mathrm{roof}} \tag{11}$$

where the first terms represent the contributions from the street canyon and the second terms those from the roofs. The values for $C_{v,\mathrm{bulk,s}}$ and $\lambda_{\mathrm{bulk,s}}$  are obtained from Equations 4 and 9  and they yield $2.5 \times 10^6\,\mathrm{J\,m^{-3}\,K^{-1}}$ and $1.53\,\mathrm{W\,m^{-1}\,K^{-1}}$, respectively.

The bulk surface thermal admittance is expressed as:

$$\mu_{\mathrm{bulk,s}} = \sqrt{\lambda_{\mathrm{bulk,s}}\,C_{v,\mathrm{bulk,s}}}. \tag{12}$$

Given the values for $C_{v,\mathrm{bulk,s}}$ and $\lambda_{\mathrm{bulk,s}}$ above, one obtains $\mu_{\mathrm{bulk,s}} = 1.96 \times 10^3\,\mathrm{Jm^{-2}K^{-1}s^{-1/2}}$. It lies within the range for the thermal admittance of the 'compact' and 'open' climate zones in Table 4 of Stewart and Oke (2012), and also within the uncertainty range obtained by De Ridder et al. (2012). Although this is not a formal validation, these correspondences give confidence to the default parameter values of $C_{v,s}$ and $\lambda_s$  above and to the enhanced effective surface heat capacity and heat conductivity expressed by Equations 4 and 9.

It needs to be noted that the presented methodology above assumes a homogeneous surface temperature of the urban canopy. This is also case for the next section with regard to the surface radiation properties. Consequentially, the scheme does not explicitly represent the temperature variety among the different elements in the urban canopy resulting from shadowing and the heterogeneous thermal and radiative properties. Therefore, urban-physical processes resulting from such variety are not explicitly resolved. This choice was made for providing consistency with the bulk urban land-surface schemes employing bulk parameters.

**2.1.2 Surface radiation**

In this section, the methodology for deriving the bulk (or effective) albedo $\alpha_{\mathrm{bulk}}$ and emissivity $\epsilon_{\mathrm{bulk}}$ from urban canopy parameters is addressed. The bulk values refer to the portions of reflected incoming short-wave radiation and emitted infra-red radiation by the urban canopy layer to the upper atmosphere, respectively. It also accounts for the modulation of the bulk value according to the increased-albedo effect of snow. The bulk albedo reduction factor of the urban canopy $\psi_{\mathrm{bulk}}$ is derived from the canyon height-to-width ratio $\frac{h}{w_c}$  and roof fraction $R$:

$$\alpha_{\mathrm{bulk}} \simeq \left((1-f_{\mathrm{snow}})\alpha + f_{\mathrm{snow}}\alpha_{\mathrm{snow}}\right)\psi_{\mathrm{bulk}}\left(\frac{h}{w_c}, R\right) \tag{13}$$

where $\alpha$ is the surface albedo and $f_{snow}$ is the snow-covered fraction. $\psi_{bulk}\left(\frac{h}{w_c}, R\right)$ is calculated by:

$$\psi_{bulk}\left(\frac{h}{w_c}, R\right) = R + (1-R)\,\psi_{canyon}\left(\frac{h}{w_c}\right) \tag{14}$$

where $\psi_{canyon}\left(\frac{h}{w_c}\right)$ is the canyon albedo reduction factor. Hereby, the total albedo reduction factor is calculated from the albedo reduction of the roof weighted with the roof fraction $R$, and that of the street-canyon weighted with the canyon fraction

5   $(1-R)$. As stated before, flat roofs are considered, hence the roof fraction $R$ does not lead to a albedo reduction. In contrast, multiple reflections take place for the street-canyon for which the canyon albedo reduction factor $\psi_{canyon}$ is taken into account.

Instead of implementing a computationally demanding explicit canyon radiation scheme, an approximation for $\psi_{canyon}$ is proposed to the numerical estimation from Fortuniak (2007). The latter applies an exact solution of the multiple-reflection problem allowing to subdivide the different facets in an urban canyon. The exact solution results in a high accuracy for low

10  solar heights when the lower canyon parts are shaded. It could reproduce the effective-albedo observations from a scale model (Aida, 1982) and from a real canyon very well. The numerical estimation shows that the albedo reduction is most sensitive to the $\frac{h}{w_c}$-ratio, hence the following approximation is proposed:

$$\psi_{canyon}\left(\frac{h}{w_c}\right) = \exp\left(-0.6\,\frac{h}{w_c}\right) \tag{15}$$

This closely matches the numerical estimation with a maximal error of $\pm 7\%$ for the highest excursion of the sun during

15  summer solstice at the mid-latitude ($55°$), a canyon parallel to the solar azimuth, and  an albedo of 0.4 (Fortuniak, 2007, see their Figure 11). With regard to other sun heights, canyon directions relative to the solar azimuth, and $\frac{h}{w_c}$-ratios between 0 and 2 (Fortuniak, 2007, see their Figures 8 and 11), the proposed $\psi_{canyon}$-formulation has a maximal error of $45\%$. It should be noted that the approximation is fitted to the numerical estimation for a perfect urban canyon. Hence, the approximation neglects additional albedo changes due to bending roofs and varying albedos for the different facets.

20  Optionally, a distinction is made between the albedo of roofs, roads and walls as follows:

$$\alpha_{bulk} \simeq \frac{\left[\alpha_{road,snow} + 2\frac{h}{w_c}\alpha_{wall,snow}\right]}{\left(1 + 2\frac{h}{w_c}\right)}\,\psi_{canyon}\left(\frac{h}{w_c}\right)(1-R) + \alpha_{roof,snow}R \tag{16}$$

with

$$\alpha_{i,snow} = (1 - f_{snow})\,\alpha_i + f_{snow}\,\alpha_{snow}, \text{ for } i \text{  in (roof, wall, road)} \tag{17}$$

and where $\frac{\left[\alpha_{road,snow} + 2\frac{h}{w_c}\alpha_{wall,snow}\right]}{\left(1 + 2\frac{h}{w_c}\right)}$ 
[revised manuscript text omitted]